# Combining Human Predictions with Model Probabilities via Confusion Matrices and Calibration

**Gavin Kerrigan**[1]     **Padhraic Smyth**[1]     **Mark Steyvers**[2]
[1]Department of Computer Science     [2]Department of Cognitive Sciences
University of California, Irvine
gavin.k@uci.edu     smyth@ics.uci.edu     mark.steyvers@uci.edu

## Abstract

An increasingly common use case for machine learning models is augmenting the abilities of human decision makers. For classification tasks where neither the human nor model are perfectly accurate, a key step in obtaining high performance is combining their individual predictions in a manner that leverages their relative strengths. In this work, we develop a set of algorithms that combine the probabilistic output of a model with the class-level output of a human. We show theoretically that the accuracy of our combination model is driven not only by the individual human and model accuracies, but also by the model's confidence. Empirical results on image classification with CIFAR-10 and a subset of ImageNet demonstrate that such human-model combinations consistently have higher accuracies than the model or human alone, and that the parameters of the combination method can be estimated effectively with as few as ten labeled datapoints.

## 1   Introduction

One of the main goals of machine learning is to develop algorithms that can operate robustly in an autonomous fashion without human supervision. However, there are many applications where hybrid human-machine approaches are likely to be a preferred mode of operation for a variety of different reasons, such as improving trust between humans and machines, and allowing for a human or a model to take over in situations where the one or the other lacks expertise [Kamar, 2016, Vaughan, 2018, Kleinberg et al., 2017, Riedl, 2019, Trouille et al., 2019, Johnson and Vera, 2019, Zahedi and Kambhampati, 2021].

The performance benefits of combining multiple predictors, rather than relying on a single predictor, have been clearly demonstrated in past work in several fields. For example, in machine learning there is a rich vein of research over the past few decades on combining models using a variety of different estimation and algorithmic approaches [Kittler et al., 1998, Dietterich, 2000, Kuncheva, 2014, Sagi and Rokach, 2018]. This existing line of work emphasizes that combinations of models that have diversity in how they make predictions can systematically outperform a single model. In parallel, in the behavioral science literature, there has been extensive prior work studying combinations of human opinions where, again, diverse combinations tend to outperform any single individual [Hong and Page, 2004, Lamberson and Page, 2012].

This naturally leads to questions about hybrid combinations of human and machine predictions, rather than just combining one type or the other. For example, one motivation for hybrid combinations is empirical evidence that human and machine classifiers do not make the same types of errors for problems such as image classification [Geirhos et al., 2020, Rosenfeld et al., 2018, Serre, 2019], i.e., they are diverse in their predictions. These ideas have begun to have impact in real-world applications, where hybrid human-machine teams have been found to be effective in areas such as crowdsourcing [Kamar et al., 2012], citizen science [Beck et al., 2018], speech transcription [Gaur et al., 2015],

35th Conference on Neural Information Processing Systems (NeurIPS 2021).

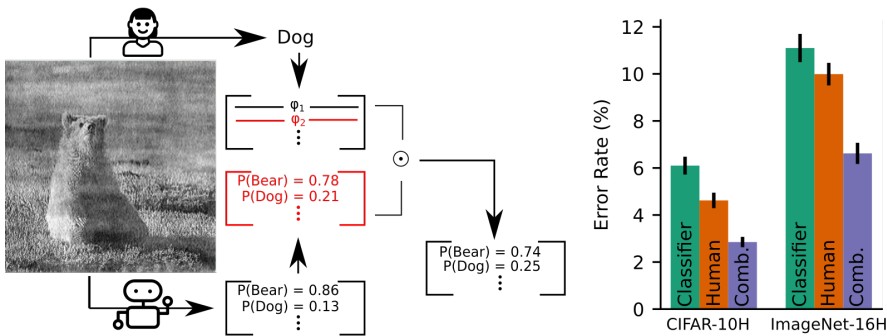

Figure 1: Left: Combining a human's label and a classifier's probabilities for an ImageNet-16H [Steyvers et al., 2022] image (true label: bear). Right: Human-machine combinations (purple) achieve lower error rates on average than the human or classifier alone (ResNet-164 on CIFAR-10H, VGG-19 on ImageNet-16H).

face identification [Phillips et al., 2018], and clinical radiology [Bien et al., 2018, Patel et al., 2019, Rajpurkar et al., 2020].

In this paper, we focus on a simple yet realistic instantiation of the general problem of hybrid combinations of human and machine predictions. In particular, we consider a $K$-way classification task such as image classification, with a single human making categorical classification decisions (i.e. predicting only a label) and a single classification model predicting a distribution over labels. While humans can provide confidence estimates with their label predictions, calibrated self-assessment of confidence can be difficult [Keren, 1991, Kahneman and Tversky, 1996, Klayman et al., 1999] as well as time-consuming.

A key question in this context is whether the non-probabilistic information from a human predictor can be effectively combined with the probabilistic information from a machine learning model. We answer this question in the affirmative and show both theoretically and empirically how relatively simple probabilistic combination techniques can robustly outperform each of a human and machine on their own. In particular we show that a human can augment their predictions with those of a classification model, improving classification accuracy and producing calibrated predictions, even when the model is less accurate than the human. Similarly, from the model's perspective, accuracy can often be significantly improved by augmenting it's class probabilities with a human's labels, while improving calibration performance, even when the human is less accurate than the model.

Figure 1 shows an example using our proposed methodology for an image from the ImageNet-16H dataset [Steyvers et al., 2022]. The human incorrectly predicts the label dog. The classification model (a deep network) predicts the correct label bear with a probability of 0.86 (uncalibrated) and 0.78 (calibrated). The combined prediction is for bear with a confidence of 0.74, with dog having a higher confidence given the human's prediction. The histograms on the right show the overall average reduction in error rate on test images: even though the classifiers are on average less accurate (6.10%, 11.1% error) than the human (4.62%, 9.99% error), the resulting combinations have lower error rates than either (2.85%, 6.62% error).

The primary contributions of our work are as follows:

- We propose and investigate a general framework for combining predictions using instance-level confidence from a model and class-level information from a human. The methods we propose are straightforward to implement in practice and label-efficient.[1]
- We empirically validate our approach on the CIFAR-10H [Peterson et al., 2019] and ImageNet-16H [Steyvers et al., 2022] image classification datasets and show that human-machine combinations in this context are systematically more accurate and better calibrated than either alone.
- We develop a theoretical understanding, for this framework, of the key tradeoffs related to calibration and accuracy for both the individual human and model, introducing the notion of model and human confidence ratios. We illustrate how these factors affect the combination, showing for

---

[1]Code for our estimation methods and experiments is available at:
https://github.com/GavinKerrigan/conf_matrix_and_calibration.

example how two models with the same accuracy but with different calibration properties can have different performance when combined with human predictions.

The paper begins in Section 2 by introducing notation and deriving our combination method. Section 3 discusses related work and in Section 4 we propose a number of different estimation methods. Section 5 describes our experimental results with two image classification datasets, with individual human labelers, individual models, and combinations. Complementing the experimental results in Section 6 we develop theoretical results that characterize combination performance. Section 7 discusses the potential societal impact and limitations of the work, and Section 8 concludes the paper.

## 2 Combining Human Labels and Model Probabilities

**Notation.** We consider a $K$-ary classification problem, where the goal is to predict a label $y \in \mathcal{Y} = \{1, \ldots, K\}$ from features $x \in \mathcal{X}$. The random variable $(x, y)$ follows an unknown distribution with support $\mathcal{X} \times \mathcal{Y}$. We assume access to an individual human labeler represented by the function $h : \mathcal{X} \to \mathcal{Y}$, where $h(x) \in \mathcal{Y}$ is the label predicted by the human. In addition, we have access to a trained machine classifier $m : \mathcal{X} \to \mathbb{R}^K$, where $m(x) \in \mathbb{R}^K$ is the normalized probability vector output by the classifier. Both the human and classifier are assumed to be noisy labelers relative to the ground truth $y$. The true labels could be determined, for example, by expert labelers or additional information not contained in $x$.

**Combining Predictions.** Given an input $x \in \mathcal{X}$, our goal is to predict a true label conditioned on the predictions $h(x)$ and $m(x)$. The key challenge in combining human and classifier predictions is simultaneously leveraging both the class-level outputs from the human and the predictive distributions output by the classifier. Although we focus on the particular case where $h$ is a human and $m$ is a classifier, our setup could be applied more generally to combinations of a non-probabilistic labeler (whose output is categorical) and a probabilistic labeler (whose output is a distribution over classes).

There are a variety of functional forms that could be used to combine the predictions. We pursue a probabilistic approach, where the conditional distribution over labels that we seek can be factored via Bayes' rule as

$$p\big(y|h(x), m(x)\big) \propto p\big(h(x)|y, m(x)\big)p\big(y|m(x)\big). \tag{1}$$

In this work, we pursue a conditional independence (CI) approach where the human labels $h(x)$ and the probabilistic predictions $m(x)$ are assumed to be conditionally independent given $y$. Under this assumption, the quantity of interest factors as

$$p\big(y|h(x), m(x)\big) \propto p\big(h(x)|y\big)p\big(y|m(x)\big). \tag{2}$$

In the above, the term $p(h(x)|y)$ can be interpreted as calibrated probabilities at the class level. We parameterize the term $p(h(x)|y)$ by the confusion matrix for the labeler $h$, which we denote by $\varphi$ with entries $\varphi_{ij} = p(h(x) = i|y = j)$. The second term $p(y|m(x))$ can be interpreted as calibrated probabilities at the instance level. However, the probabilistic output of the classifier $m(x)$ may differ from $p(y|m(x))$. For example, modern neural networks tend to be overconfident in their predictions [Guo et al., 2017]. To remedy this, post-hoc calibration maps $m(x)$ to well-calibrated probabilities via a learned calibration map with parameters $\theta$. We note that there are many possible choices for such a post-hoc calibration map [Guo et al., 2017, Zhang et al., 2020, Kull et al., 2019, Patel et al., 2021], but our method is agnostic to this choice. In this work, we use $m^\theta(x)$ to denote the output of the classifier after applying such a calibration map. The second term in Equation (2) is then parameterized by the calibrated classifier probabilities $m^\theta(x)$.

Altogether, our method expresses the predicted probability for class $j$, given that the human predicts class $i$ and the model produces a class probability vector $m(x)$ as:

$$p\big(y = j|h(x) = i, m(x)\big) = \frac{\varphi_{ij}m_j^\theta(x)}{\sum_{k=1}^K \varphi_{ik}m_k^\theta(x)}. \tag{3}$$

The CI assumption above is common (both implicitly and explicitly) in prior work on combining predictions, such as additive classifier ensembles [Kuncheva, 2014, Sagi and Rokach, 2018] and (log-)linear opinion pools [Genest and Zidek, 1986, Jacobs, 1995]. As our primary motivation is to develop a relatively simple and robust methodology for combining human and model predictions, the

additional functional or parametric assumptions (and parameters) required to specify a joint model for $p(h(x), m(x)|y)$ are beyond the scope of the work in this paper. In addition, although the CI assumption is unlikely to hold exactly, prior work [Kuncheva, 2006] notes that a CI model can be an optimal discriminant even when the CI assumption is violated. As further motivation, for the two datasets we use in this paper, CIFAR-10H and ImageNet-16H, the conditional dependence of the predictions $h(x)$ by the human and $m(x)$ by the model appears to be relatively weak (see Appendix A for details).

# 3 Related Work

We summarize below relevant aspects of related literature. While there is a significant amount of prior work in machine learning and related fields on combining predictors, this work has in general not addressed the specific problem of combining hard label predictions from a human with probabilistic label predictions from a model.

**Ensembles and Opinion Pools.** There is a rich literature in machine learning on studying predictions based on ensembles of classification models. For non-probabilistic classifiers, the most common aggregation methods are variants of (weighted) majority voting [Dietterich, 2000, Sagi and Rokach, 2018]. However, in our case of only two predictors, a weighted majority vote ensemble can never improve accuracy over the best of its two components. Beyond majority voting, naive Bayes aggregation [Xu et al., 1992, Kuncheva, 2014] fits a class-level confusion matrix to each predictor. Kim and Ghahramani [2012] develop a fully Bayesian extension of this, which relaxes the independence assumption by explicitly modeling correlations between predictors. However, because these confusion-matrix aggregation methods are at the class level they are unable to take full advantage of the instance-level uncertainties produced by the probabilistic labeler.

In the context of aggregating predictions from multiple humans, there has been a considerable amount of prior work in the behavioral sciences and forecasting literature. Approaches include additive linear and log-linear opinion pools for subjective distributions [Genest and Zidek, 1986, Jacobs, 1995], techniques for weighting linear combinations of real-valued human predictions [Lamberson and Page, 2012, Davis-Stober et al., 2015], and voting methods for combining label predictions from more than two human predictors [Lee and Lee, 2017]. A key difference between these methods and our work is that they do not address the problem of how to combine probabilistic and non-probabilistic predictions in a human-machine context.

**Leveraging Human and Model Predictions.** Combining human predictions with model predictions to solve classification problems has also been a topic of recent interest in a number of different areas. For example, in [Wright et al., 2017] simple averaging is used to combine the labels of multiple human annotators with the output of a classifier for astronomical image classification, achieving better performance than with either the humans or the classifier. In crowd-sourcing, classification models have been used to automatically filter examples to improve human annotation efficiency [Kamar et al., 2012, Russakovsky et al., 2015, Vaughan, 2018, Trouille et al., 2019]. A similar line of research focuses on algorithmic deferral techniques where a model defers to human predictions based on the model's confidence [Madras et al., 2018, Raghu et al., 2019], as well as work on adapting prediction models to the human decision maker [Branson et al., 2017, Mozannar and Sontag, 2020, Wilder et al., 2020, Bansal et al., 2021, Okati et al., 2021]. The results in [Mozannar and Sontag, 2020] in particular describe experiments with the same CIFAR-10H dataset that we use in this paper. However, in addition to being different to our work in terms of its focus on deferral (rather than combining) we also note that in [Mozannar and Sontag, 2020] the improvements in performance are demonstrated using relatively large numbers of human labels. In contrast, as we demonstrate in Section 5 on the CIFAR-10H and ImageNet-16H datasets, our methods require only a small number of human labels to yield combined predictions that are more accurate than either human or model alone. In general, existing work on filtering and deferral strategies complements the combining methods that we develop in this paper. All of these approaches are potentially useful in a broad range of human-AI applications, but in different contexts.

# 4 Estimation Methods and Algorithms

Combining human and machine predictions via Equation (3) requires learning two sets of parameters: confusion matrix parameters for the human and calibration parameters for the classifier. The choice of procedure used to infer these parameters impacts the label efficiency and quality of the resulting combination. In this section, we describe in detail several inference procedures and empirically evaluate them in the context of human-machine combinations.

To estimate our combination model, we assume access to a combination dataset $\mathcal{D}_C = \{(h(x_\ell), m(x_\ell), y_\ell)\}_{\ell=1}^n$ with human labels, machine probabilities, and ground truth labels. We assume the machine classifier is already pre-trained using true labels on a separate training set $\mathcal{D}_T$.

**Confusion Matrix Estimation.** Recall that $\varphi$ denotes a $K \times K$ confusion matrix for the human, where $\varphi_{ij}$ is $p(h(x) = i|y = j)$. The most straightforward estimate for this quantity is the maximum likelihood estimate, where $\varphi_{ij}$ is estimated by the number of datapoints in $\mathcal{D}_C$ where the human labeler predicts $h(x) = i$ when the ground truth is $y = j$, normalized by the number of points in $\mathcal{D}_C$ where $y = j$.

However, as the size of the confusion matrix is quadratic in $K$, this estimate will have high variance for small amounts of labeled data and collecting enough human labels to overcome this variance could be prohibitively expensive. Moreover, such a matrix depends on the particular labeler, and would potentially need to be re-estimated for different individual human labelers. We can instead take a Bayesian approach and incorporate informative prior information. Given the true label $y = j$, the human label $h|y = j \sim \text{Discrete}(\varphi_{*j})$ is assumed to be drawn from a discrete distribution with parameters corresponding to the $j$th row in the confusion matrix. We place a conjugate Dirichlet prior $\varphi_{*j} \sim \text{Dirichlet}(\alpha_j)$ over each column with parameters $\alpha_j \in \mathbb{R}^k$. The prior parameters $\alpha_j$ are chosen such that

$$(\alpha_j)_i = \begin{cases} \beta & i \neq j \\ \gamma & i = j \end{cases} \tag{4}$$

That is, the prior matrix is $\gamma \in \mathbb{R}_{>0}$ along the diagonal and $\beta \in \mathbb{R}_{>0}$ on the off-diagonal. We choose $\beta$ and $\gamma$ such that the resulting Dirichlet distribution has mode equal to the train-set accuracy of the classifier, which can be obtained from the training data without needing additional human labels. This choice of prior reflects our belief that the confusion matrix will have a diagonally dominant structure, and moreover corresponds to a prior belief that the human is equally accurate across all classes. Posterior estimates of the confusion matrix can then be obtained straightforwardly by conjugacy.

**Calibration Parameter Estimation.** Scaling-based calibration maps are typically fit by optimizing the log-likelihood on a held-out calibration set [Guo et al., 2017, Zhang et al., 2020, Kull et al., 2019]. In this section, we describe a Bayesian version of temperature scaling [Guo et al., 2017], allowing us to incorporate informative prior information. A temperature $T \in (0, 1)$ indicates underconfidence, and $T \in (1, \infty)$ indicates overconfidence. To account for this difference in scale, we place a Gaussian prior on the log-temperature $\log T = \tau \sim \mathcal{N}(\mu, \sigma^2)$. As this is a non-conjugate prior, the maximum a posteriori (MAP) temperature is estimated via gradient-based optimization. In our experiments, we choose $\sigma = 0.5$ for the CIFAR-10 models and $\sigma = 0.75$ for the ImageNet models, and we use $\mu = 0.5$ throughout. These parameters were chosen to reflect our belief that deep models tend to be overconfident and to concentrate the prior on reasonable temperature values. We choose a larger value of $\sigma$ (i.e. a wider prior) for the ImageNet models, as this task is more difficult than CIFAR-10 and hence could potentially lead to a stronger degree of miscalibration. In Appendix B, we derive a fully Bayesian approach where we marginalize over the posterior distribution over temperature (e.g. using Monte Carlo methods [Hoffman and Gelman, 2014]). However, we find empirically that the simpler MAP approach is more effective and, as a result, focus on MAP estimation in this paper.

**Learning without Ground Truth.** Requiring both human and ground truth labels in $\mathcal{D}_C$ can be a potentially limiting assumption in domains where ground truth labels are unavailable or expensive to obtain. To avoid this, we propose an unsupervised approach that is able to learn both calibration parameters and confusion matrix parameters from a combination dataset of the form $\mathcal{D}_C = \{(h(x_\ell), m(x_\ell))\}$, consisting only of human labels and machine probabilities. We treat the ground truth labels as latent and fit the required parameters using Expectation-Maximization (EM)

Table 1: Summary of combination methods studied in this work. Except for logistic regression, parameter counts correspond to calibration using MAP temperature scaling (one parameter), and confusion matrices are fit with MAP inference. The human output is always a label.

| Method Name | Acronym | Parameters | Model Output | Ground Truth? | Label Efficient? |
|---|---|---|---|---|---|
| Logistic Regression | LR | $k^2 + 2k$ | Probabilities (P) | ✓ | ✗ |
| Calibrated Machine Probs & Single-Parameter Confusion | SP | 2 | Probabilities (P) | ✓ | ✓ |
| Machine Labels & Human Labels | L+L | $2k^2$ | Labels (L) | ✓ | ✗ |
| Calibrated Machine Probs. & Human Labels | P+L | $k^2 + 1$ | Probabilities (P) | ✓ | ✓ |
| Calibrated Machine Probs. & Human Labels (EM) | P+L-EM | $k^2 + 1$ | Probabilities (P) | ✗ | ✓ |

[Dempster et al., 1977] (details in Appendix C). This approach can be seen as an extension of the Dawid-Skene model [Dawid and Skene, 1979]. In particular, the Dawid-Skene model fits a confusion matrix for the classifier, whereas in contrast our work fits calibration parameters for the classifier. We perform both maximum likelihood and MAP estimation with this method, using the same priors as above.

For clarity of exposition, we focus on three types of combinations in our results:

- **Machine Labels & Human Labels (L+L)**, a baseline where the instance-level probabilities from the model are discarded and instead a confusion matrix is fit for the model, as well as a confusion matrix for the human. The confusion matrices are estimated via supervised MAP inference. This can be viewed as a naive Bayes' combination [Kuncheva, 2014, Chapter 4] for non-probabilistic predictors.
- **Calibrated Machine Probabilities & Human Labels (P+L)** combined via Equation (3) using *supervised* MAP estimates for both the calibration parameters and confusion matrix parameters.
- **Calibrated Machine Probabilities & Human Labels (P+L-EM)** combined via Equation (3) using *unsupervised* MAP estimates fit with our EM algorithm, i.e. using human labels but no ground truth in correspondence with these labels.

In terms of complexity, these methods sit between a simple model with only one parameter for the human confusion matrix (**SP**) (where the diagonal entry corresponds to the human's marginal accuracy), and a full multinomial logistic regression model (**LR**).[2] We find that **LR** can obtain slightly lower error rates in some cases, but requires significantly more labeled data than the other methods to do so. At the other extreme, **SP** is highly data efficient, but underfits compared to our preferred methods. We provide additional discussion of these methods to Appendix E. The various estimation methods discussed in this section are summarized in Table 1.

## 5   Experiments

**Datasets and Models.**   We evaluate various combination strategies on two pre-existing image classification datasets that include human annotations: CIFAR-10H [Peterson et al., 2019] and ImageNet-16H [Steyvers et al., 2022].[3] CIFAR-10H contains 10-way human classifications for 10,000 images from the standard CIFAR10 test set [Krizhevsky et al., 2009]. ImageNet-16H contains 16-way human classifications for noisy images from the ImageNet test set [Deng et al., 2009], distorted by phase noise at each spatial frequency based on four levels of phase noise (80, 95, 110, and 125). The human classifications for CIFAR-10H and ImageNet-16H come from the Amazon Mechanical Turk platform.

We experiment with four CNN models on CIFAR-10H (ResNet-110, Resnet-164 [He et al., 2016a], PreResNet-164 [He et al., 2016b], DenseNet [Huang et al., 2017]), and eight models (four VGG-19 [Simonyan and Zisserman, 2015], four GoogLeNet [Szegedy et al., 2015]) of varying accuracy on ImageNet-16H. Our models are chosen to span a range of performance, from below human accuracy to exceeding human accuracy. See Appendix F for details regarding our model architectures and training procedures.

---

[2]In fact, the CI combination with temperature scaling can be seen as a special case of multinomial logistic regression taking $m(x)$ and $h(x)$ as inputs (see Appendix D).

[3]The ImageNet-16H dataset is available on the Open Science Foundation at https://osf.io/2ntrf/.

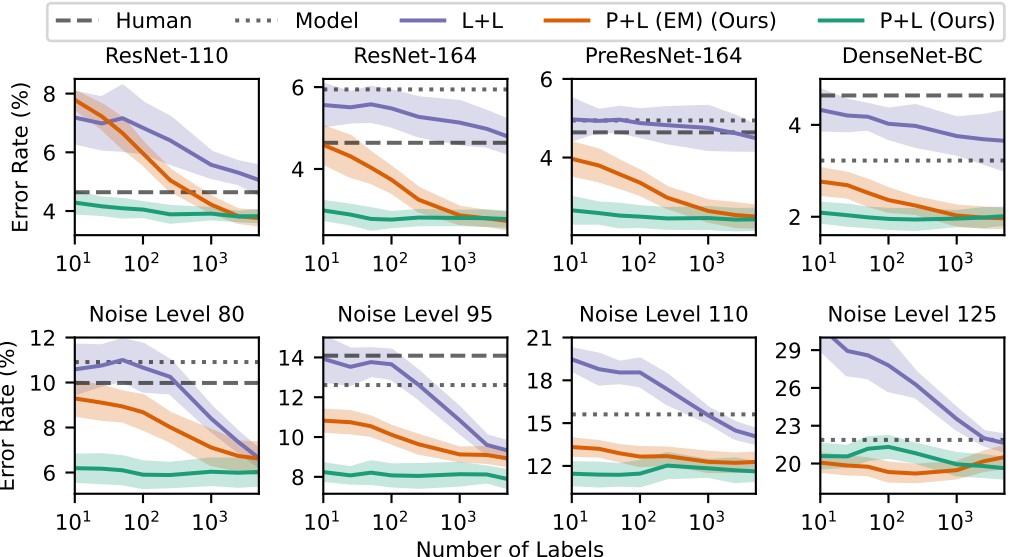

Figure 2: Learning curves for various models on CIFAR-10H (top) and VGG-19 on ImageNet-16H (bottom). For the supervised methods (L+L, P+L), the x-axis corresponds to the number of human labels, machine predictions, and ground truth labels. For the unsupervised method (P+L (EM)), the x-axis corresponds to the number of human labels and machine predictions (but no ground truth). In all cases, the y-axis is the error rate, with shading indicating one standard deviation.

Both datasets are partitioned into three disjoint subsets: (i) a model training set $\mathcal{D}_T$, (ii) a combination training set $\mathcal{D}_C$, and (iii) an evaluation set $\mathcal{D}_E$. The model training set $\mathcal{D}_T$ is the same as the suggested training set for the original CIFAR-10 and ImageNet datasets, and is used to fit the classification models. The combination training set $\mathcal{D}_C$ is used to estimate any calibration parameters and confusion matrices, and the held-out evaluation set $\mathcal{D}_E$ is used solely for testing. The combination training set and evaluation set are subsets of the original test sets, where 70% of the data is used for fitting the combinations and 30% is used for evaluation. The true labels (ground truth) for both CIFAR-10 and ImageNet correspond to the originally-provided labels for each of these datasets. In our experiments, (i) is fixed and we average over randomly selected splits for (ii) and (iii).

Importantly, we note that each individual human only labels a small subset of the images in each dataset (approximately 200 images per individual on CIFAR-10H, and 50 images per noise level and individual on ImageNet-16H). To overcome this limitation in our experiments, we select a single human-generated label for each image by random sampling. This sampled label is fixed across all trials. In Appendix G, we demonstrate that our results are robust to this sampling procedure by fitting a Bayesian P+L combination to each individual human (i.e. without the aforementioned sampling procedure). In short, this results in human-model combinations with performance on par with that obtained by sampling.

**Calibration Methods.** In our experiments, model calibration is carried out by MAP temperature scaling (TS) [Guo et al., 2017]. In Appendix H, we experiment with two additional calibration methods: ensemble temperature scaling [Zhang et al., 2020] and I-Max binning [Patel et al., 2021]. We find that the combination performance is robust to the choice of calibration map, and hence restrict our focus to TS in this section. In addition, we find that combinations using uncalibrated model probabilities produce less accurate combinations than those using calibrated model probabilities.

**Learning Curves.** Given that it is highly desirable to learn human-machine combinations from small amounts of data, we empirically study the data efficiency of the various inference methods previously described. In Figure 2, we plot the combination error rate on the evaluation data as a function of dataset size for the four CIFAR-10 models (first row) and the VGG-19 model on ImageNet (second row). For supervised methods, the dataset size corresponds to the number of $(h(x), m(x), y)$

| Metric | Model Name | No Calibration | | 10 Datapoints | | 5000 Datapoints | |
|---|---|---|---|---|---|---|---|
| | | Model | Comb. | Model | Comb. | Model | Comb. |
| ECE ($10^{-2}$) | ResNet-110 | $5.23 \pm 0.35$ | $2.08 \pm 0.25$ | $3.03 \pm 0.58$ | $1.30 \pm 0.23$ | $2.99 \pm 0.36$ | $1.76 \pm 0.18$ |
| | ResNet-164 | $2.98 \pm 0.34$ | $1.63 \pm 0.23$ | $1.95 \pm 0.33$ | $1.25 \pm 0.18$ | $1.89 \pm 0.32$ | $1.39 \pm 0.18$ |
| | PreResNet-164 | $3.03 \pm 0.29$ | $1.87 \pm 0.22$ | $2.31 \pm 0.33$ | $1.40 \pm 0.26$ | $2.27 \pm 0.31$ | $1.43 \pm 0.21$ |
| | DenseNet-BC | $2.18 \pm 0.27$ | $1.53 \pm 0.20$ | $1.76 \pm 0.28$ | $1.34 \pm 0.14$ | $1.73 \pm 0.28$ | $1.27 \pm 0.13$ |
| cwECE ($10^{-2}$) | ResNet-110 | $0.81 \pm 0.07$ | $0.23 \pm 0.05$ | $0.58 \pm 0.07$ | $0.24 \pm 0.05$ | $0.58 \pm 0.06$ | $0.19 \pm 0.06$ |
| | ResNet-164 | $0.39 \pm 0.06$ | $0.15 \pm 0.03$ | $0.31 \pm 0.05$ | $0.15 \pm 0.04$ | $0.31 \pm 0.05$ | $0.13 \pm 0.03$ |
| | PreResNet-164 | $0.29 \pm 0.04$ | $0.13 \pm 0.03$ | $0.28 \pm 0.04$ | $0.13 \pm 0.03$ | $0.28 \pm 0.04$ | $0.13 \pm 0.03$ |
| | DenseNet-BC | $0.23 \pm 0.03$ | $0.11 \pm 0.02$ | $0.24 \pm 0.02$ | $0.12 \pm 0.02$ | $0.24 \pm 0.02$ | $0.11 \pm 0.02$ |
| NLL | ResNet-110 | $0.40 \pm 0.02$ | $0.16 \pm 0.01$ | $0.35 \pm 0.02$ | $0.15 \pm 0.01$ | $0.35 \pm 0.02$ | $0.14 \pm 0.01$ |
| | ResNet-164 | $0.24 \pm 0.02$ | $0.11 \pm 0.01$ | $0.20 \pm 0.01$ | $0.10 \pm 0.01$ | $0.20 \pm 0.01$ | $0.10 \pm 0.01$ |
| | PreResNet-164 | $0.23 \pm 0.02$ | $0.13 \pm 0.02$ | $0.19 \pm 0.02$ | $0.11 \pm 0.01$ | $0.19 \pm 0.02$ | $0.10 \pm 0.01$ |
| | DenseNet-BC | $0.17 \pm 0.01$ | $0.10 \pm 0.01$ | $0.14 \pm 0.01$ | $0.09 \pm 0.01$ | $0.14 \pm 0.01$ | $0.08 \pm 0.01$ |

Table 2: Calibration metrics for various CIFAR-10H models with P+L combinations. Even a small amount of labeled data (10 labels) reduces the calibration error of both the classifier and combination. For all metrics, lower is better.

triples used for learning, whereas for unsupervised methods the dataset size corresponds to the number of $(h(x), m(x))$ pairs without ground truth $y$.

As Figure 2 demonstrates, the P+L method is able to learn a human-model combination that outperforms both the human and model alone, with very few datapoints. While the P+L-EM method requires more human labels than the P+L method, it is able to learn such a combination without any ground-truth labels. The baseline L+L method fails to learn an effective combination on the CIFAR-10 dataset, and only does so on the VGG-19 ImageNet dataset with a large number of ground truth labeled datapoints. This demonstrates that the instance level probabilities from the model are a key component in efficiently learning human-model combinations with high accuracy. In Appendix E, we provide similar plots for GoogLeNet and for maximum likelihood based inference procedures. In general, we find that maximum likelihood estimation requires more data than MAP estimation, and hence we focus our presentation of results on MAP.

**Calibration Properties of Combinations.** In addition to the error rate, we study the calibration properties of P+L combinations. In Table 2, we report the ECE [Guo et al., 2017], classwise-ECE (cwECE) [Kull et al., 2019, Patel et al., 2021], and negative log-likelihood (NLL) for our various CIFAR-10 models (Model) and the resulting human-machine combinations (Comb.) on the held-out evaluation set. The ECE and cwECE are evaluated using 15 bins containing an equal number of data points. In addition to P+L combinations fit with 10 or 5000 labeled datapoints, we evaluate a combination consisting of the uncalibrated classifier probabilities with the MAP human confusion matrix estimated with 5000 labeled datapoints (No Calibration).

Combining classifier probabilities with human labels generally results in a combination that is better calibrated than the model alone. Moreover, MAP TS can be fit using a very small number of labeled datapoints. Our results show that the calibration properties (of both the classifier alone and the resulting human-machine combination) significantly improve with only ten labeled examples. However, increasing the number of labeled examples to 5000 does not result in further calibration gains. We provide similar results for our ImageNet-16H models in Appendix I.

## 6 Theoretical Analysis

**Confidence Ratios.** The key quantity in our analysis is the *confidence ratio* of a predictor, which is a random variable representing a predictor's confidence for the correct class relative to the predictor's confidence for other classes. This quantity can be thought of as the predictor's instance-level odds for making a correct prediction. More specifically, the confidence ratios $r_m$ and $r_h$ for machine and human labelers are defined as

$$r_m(x) = \frac{m_y^\theta(x)}{1 - m_y^\theta(x)} \qquad r_h(x) = \frac{\varphi_{h(x)y}}{1 - \varphi_{h(x)y}} \tag{5}$$

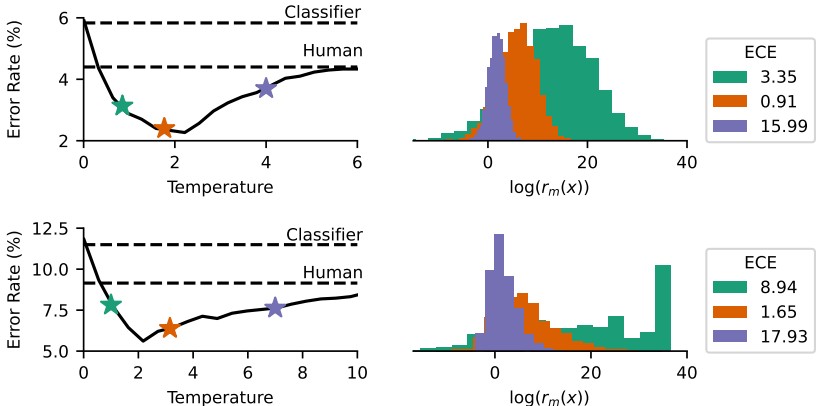

Figure 3: Left: Single human labeler combined with various classifiers of equal accuracy, resulting in combinations with varied accuracies. The orange point corresponds to the P+L combination. First row: ResNet-164 on CIFAR-10H. Second row: VGG-19 on ImageNet-16H at noise level 80.

We note that unlike the machine classifier, the human does not directly output such confidences – rather, this quantity is estimated empirically through the human's confusion matrix.

If the model has a confidence ratio of $r_m(x) > 1$ (indicating that the model has confidence greater than $0.5$ for the correct class), then the model is guaranteed to correctly label $x$. On the other hand, $r_m(x) > 1$ is not sufficient for the combination to correctly label $x$ – instead, the model must be sufficiently confident in its prediction as well. The following theorem formalizes this notion by a lower bound on the accuracy of the combination in terms of the confidence ratios of the individual predictors. For binary classification tasks, this lower bound is achieved, i.e. we have equality.

**Theorem 1** (Combination Model Accuracy.). *The accuracy of the P+L combination $c(x)$ is at least the probability that the confidence ratio for $m$ exceeds the inverse confidence ratio for $h$.*

$$\mathbb{E}\left[\mathbb{1}\left(c(x) = y\right)\right] \geq p(r_m(x) \geq (r_h(x))^{-1}) \tag{6}$$

A detailed proof is provided in Appendix J. An analogous result holds for the combination of two probabilistic classifiers or two non-probabilistic classifiers. As our focus is on combining human predictions with model probabilities, we discuss these cases in Appendix J.

Theorem (1) is further illustrated in Figure 3 for a ResNet-164 classifier on CIFAR-10H (first row) and a VGG-19 classifier on ImageNet-16H (second row). For each row, we create a family of classification models where each model makes the same class-level predictions (and hence has the same error rate) but with different confidence ratio distributions. This is achieved by tempering the output probabilities of a base classifier via the map $m(X) \mapsto (m_1(X)^{1/T}, \ldots, m_k(X)^{1/T})/\sum_{i=1}^{K} m_i(X)^{1/T}$ with temperature $T > 0$ [Guo et al., 2017, Zhang et al., 2020]. In the first column of each row, the solid curve plots the error rate of the combination of a fixed human with the various classification models. Despite each classification model having the same accuracy, the accuracy of the resulting human-model combination varies.

This behavior can be explained by Theorem (1), which tells us that the combination accuracy is driven not only by the human and machine accuracies but by their confidence ratios as well. At large temperatures (purple), the classifier becomes underconfident in its predictions, and the combination error rate approaches the human error rate. At small temperatures (green), the model becomes overconfident in its predictions, and the combination error rate approaches the model error rate. When the combination is fit by our P+L method (orange), the classifier is well-calibrated (reflected by its low ECE), and the resulting combination obtains a lower error rate than each of the under- and over-confident classifier combinations.

**Relationship between combination and calibration/confusion error.** We additionally quantify the estimation error in our P+L method, incurred by empirically estimating the human confusion

matrix and calibration parameters. The result below shows we can upper bound our estimation error by the estimation error for the confusion matrix and the $\ell_1$ marginal calibration error (MCE) [Kumar et al., 2019].

**Theorem 2** (Estimation Error Upper Bound.). *Let $\eta(x,y) = |p(h(x)|y)p(y|m(x)) - m_y^\theta(x)\widehat{\varphi}_{h(x)y}|$ be the estimation error (up to normalizing constants), where $\widehat{\varphi}_{ij}$ represents an estimate of $p(h(x) = i|y = j)$. Under the CI assumption, in expectation over random $(x,y)$ pairs,*

$$\mathbb{E}_{(X,Y)\sim\mathcal{D}}\left[\eta(X,Y)\right] \leq ||\varphi - \widehat{\varphi}||_1 + MCE(m^\theta) \tag{7}$$

(Proof in Appendix J). With a sufficient amount of labeled data, the confusion matrix error $||\varphi - \widehat{\varphi}||_1$ can be made arbitrarily small (i.e. the MAP confusion matrix estimate is unbiased). This is not necessarily the case for MCE$(m^\theta)$. However, if an asymptotically unbiased calibration method is used, this result guarantees that our posterior estimation error will converge to zero.

# 7 Limitations and Potential Societal Impacts

**Limitations.** One limitation of our work is that our experiments only involve two datasets and both involve image classification. Thus, there is no guarantee that similar results (in terms of combined improvements) are achievable for other tasks, such as question-answering from text data or more general problem-solving tasks. Another potential limitation is our reliance on conditional independence in our approach. A reverse view of this is that both our theoretical and experimental results demonstrate that there is ample room for improving human and machine performance by combining their predictions, even without taking dependence into account.

In Appendix K, we find that combining a probabilistic model and a non-probabilistic model (rather than a human) often does not improve over the better of the two. We hypothesize that this is due to a lack of diversity in the predictions of two models. Precisely characterizing the dependence of our method on the predictor diversity is an interesting direction for future work, as well as developing methods for combining probabilistic and non-probabilistic predictions that are robust to this lack of diversity.

**Potential Societal Impacts.** Combining human and machine predictions to improve overall classification accuracy has the potential for positive societal benefit, particularly for example in high-stakes applications such as medical image diagnosis and autonomous driving. However, there are also potential negative societal impacts. For example, if there is a lack of transparency in terms of how the system operates (e.g., how predictions are being combined to arrive at a final result), augmenting an individual's predictions with a machine's could have negative psychological consequences for the individual, such as decreasing trust, reducing individual autonomy, and eventual disengagement.

# 8 Conclusions

We investigate methods for combining predictions using instance-level confidence from a model and class-level information from a human. Across a variety of image classification experiments our proposed combination framework leads to systematic increases in accuracy over both the model and human alone while requiring few human labels. Supporting theory illustrates how combined human-model performance is affected by calibration properties of the model.

## Acknowledgements and Funding Disclosure

We thank Heliodoro Tejeda for contributing trained models to this project. This material is based upon work supported in part by the HPI Research Center in Machine Learning and Data Science at UC Irvine, by the National Science Foundation under grants number 1900644 and 1927245, by the Center for Statistics and Applications in Forensic Evidence (CSAFE) through Cooperative Agreements 70NANB15H176 and 70NANB20H019 between NIST and Iowa State University, which includes activities carried out at the University of California Irvine, and by a Qualcomm Faculty Award. Additional revenues potentially related to this work include: research funding from NIH, PCORI, DARPA, and SAP; consulting income from Amazon.com.

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
