# Appendix A   Assessing Conditional Independence/Dependence in CIFAR-10H and Imagenet-16H Datasets

We investigate the degree to which our conditional independence assumption is satisfied empirically in the datasets used in the paper. Specifically, of interest is the assumption of conditional independence of $m(x)$ and $h(x)$, given $y$. Assessing conditional independence is not straightforward given that $m(x)$ is a $K$-dimensional real-valued vector and $h(x)$ and $y$ each take one of $K$ categorical values, with $K = 10$ for CIFAR-10H and $K = 16$ for ImageNet-16H. While there exist statistical tests for assessing conditional independence for categorical random variables, with real-valued variables the situation is less straightforward and there are multiple options such as different non-parametric tests involving different tradeoffs [Runge, 2018, Marx and Vreeken, 2019, Mukherjee et al., 2020, Berrett et al., 2020].

Given these issues we investigate the degree of conditional dependence using two relatively simple approaches. The first approach looks at the conditional mutual information (CMI) between the predicted label from the model and the predicted label from the human, conditioned on the true label. While this is indirect, in that it does not use the real-valued scores, it does allow us to measure CMI in a straightforward manner given that all the variables involved are categorical. The CMI is defined as

$$\mathrm{CMI}(M; H|Y) \;=\; \sum_y p(y) \sum_{m,h} p(m, h|y) \log \frac{p(m, h|y)}{p(m|y)p(h|y)}$$

where $M, H, Y$ are the $K$-ary random variables for the model, human, and true labels respectively (taking values $m, h, y$). The inner sum over $m, h$ is the mutual information between $M$ and $H$ conditioned on a particular value of $Y = y$. All probabilities were estimated using relative frequencies (maximum likelihood) from the evaluation sets for each dataset.

Table 3 shows the results for the 4 different models for CIFAR-10H and the $2 \times 4$ different combinations of models and noise for ImageNet-16H. To put the CMI numbers on an interpretable scale, we also compute the (unconditional) mutual information between $M$ and $H$ in each case. If $M$ and $H$ are truly independent conditioned on $Y$, then the true CMI values should be 0.

The broad conclusion from Table 3 is that for the CIFAR-10H there appears to be little to no conditional dependence (of model labels and human labels, given true labels) given that the CMI values are very close to 0. For the ImageNet-16H data the CMI values are higher, suggesting evidence for weak conditional dependence in this dataset, particularly at high noise levels where neither the human or the model are very accurate.

Figures 4, 5, 6 show the results of another assessment, now using model probabilities, for the 4 models for the CIFAR-10H data, for VGG-19 on ImageNet-16H, and for GoogLeNet on ImageNet-16H, respectively. The x-axis in each plot is the mean probability from the model for the true label $y$, conditioned on $Y = y$. The y-axis shows the mean probability (in red) from the model for the true

Table 3: Conditional and unconditional mutual information for various datasets and models.

| Dataset | Model | Noise | CMI$(M; H|Y)$ | MI$(M; H)$ |
|---|---|---|---|---|
| CIFAR-10H | DenseNet | | 0.030 | 2.829 |
| CIFAR-10H | PreResNet-164 | | 0.043 | 2.770 |
| CIFAR-10H | ResNet-110 | | 0.037 | 2.404 |
| CIFAR-10H | ResNet-164 | | 0.038 | 2.707 |
| ImageNet-16H | VGG-19 | 80 | 0.119 | 2.954 |
| ImageNet-16H | VGG-19 | 95 | 0.174 | 2.816 |
| ImageNet-16H | VGG-19 | 110 | 0.230 | 2.277 |
| ImageNet-16H | VGG-19 | 125 | 0.314 | 1.527 |
| ImageNet-16H | GoogLeNet | 80 | 0.121 | 2.825 |
| ImageNet-16H | GoogLeNet | 90 | 0.161 | 2.643 |
| ImageNet-16H | GoogLeNet | 110 | 0.260 | 2.182 |
| ImageNet-16H | GoogLeNet | 125 | 0.364 | 1.421 |

label $y$, conditioned now on **both $Y = y$ and $H = y$**, i.e., conditioned on the event that the human also predicts the true label.

If the model's probabilities for the true labels are independent of $H = y$, then the x and y values should be the same (i.e., on the diagonal). The degree to which these points (in red) are not on the diagonal is an indication of some conditional dependence of the model's probabilities on the human labels $h$. The red points are generally close to the diagonal, or slightly above (indicating, not surprisingly, that if the human predicts the true label, the model's probability for the true label tends to increase slightly (if at all) rather than decrease.) To put these values on an appropriate scale we also compute (empirically from the data) the maximum possible increase that could occur, when additionally conditioning on the human label $h$ being correct (the black points). The conclusions are similar to what we found with conditional mutual information, namely, that there is little indication of conditional dependence in the CIFAR-10H data, and some indication of dependence in the ImageNet-16H data, particularly for higher noise levels.

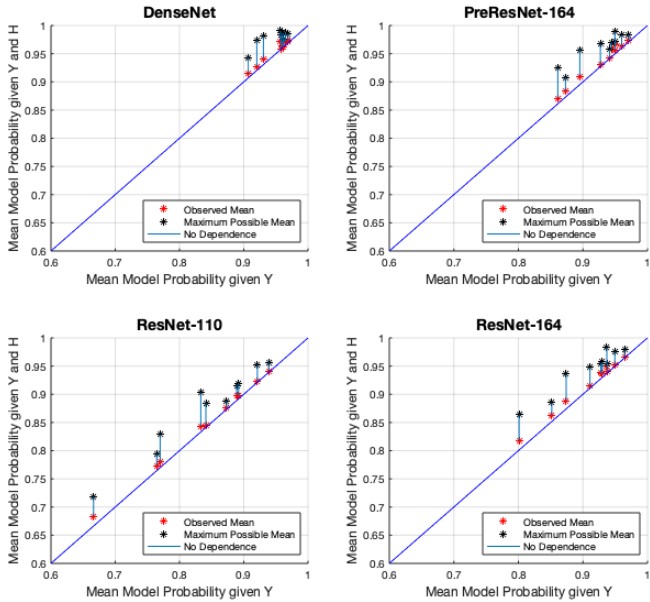

Figure 4: Change in expected values of model probabilities on CIFAR-10H data for the true class $y$, conditioning on just $y$ (x-axis), versus conditioning on both $y$ and $h(x) = y$ (y-axis, in red).

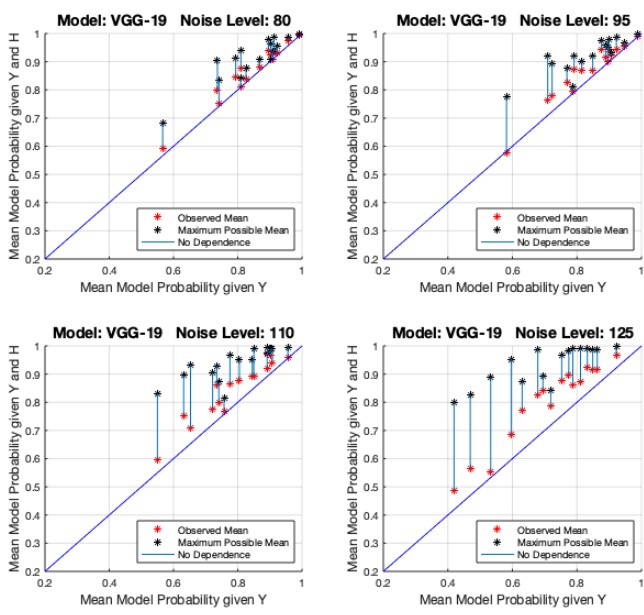

Figure 5: Same as Figure 4 but for VGG-19 models on ImageNet-16H data.

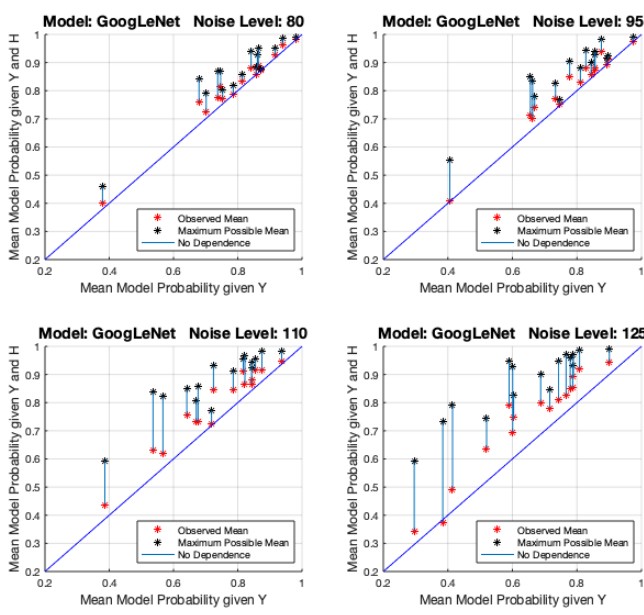

Figure 6: Same as Figure 4 but for GoogLeNet models on ImageNet-16H data.

# Appendix B   Derivation of Fully Bayesian Model for TS

In this section, we derive a fully Bayesian method for combining classifier probabilities with human labels. In summary, we place a Gaussian prior on the log-temperature (for calibration) and independent Dirichlet priors over the columns of the human confusion matrix. The posterior human confusion matrix is available in closed-form (due to conjugacy), and we sample from the posterior distribution over calibration parameters using MCMC. To predict on a new datapoint, we marginalize over the calibration and confusion parameters using the sampled temperatures and closed-form posterior confusion parameters. This marginalization is only approximate due to the required sampling step.

In more detail, let $\varphi_{*i} \sim \text{Dirichlet}(\alpha_i)$ for $i = 1, 2, \ldots, k$ be priors over the columns of the confusion matrix, and let $\log T = \tau \sim \mathcal{N}(\mu_0, \sigma_0^2)$ be a prior over the log-temperature. We use $\varphi$ to denote the confusion matrix with columns $\varphi_{*1}, \ldots, \varphi_{*K}$. We assume a fully labeled dataset is available, and of the form $\mathcal{D} = \{(h_\ell, m_\ell, y_\ell)\}$. Take the calibration and confusion parameters to be conditionally independent given the data:

$$p(\tau, \varphi | \mathcal{D}) = p(\tau | \varphi, \mathcal{D}) p(\varphi | \mathcal{D}) = p(\tau | \mathcal{D}) p(\varphi | \mathcal{D}) \tag{8}$$

The confusion parameters have a conjugate prior, but the calibration parameters do not – hence, suppose that we have sampled $\{\tau_1, \ldots, \tau_{n_s}\}$ from the posterior $p(\tau | \mathcal{D})$. To do inference on a new datapoint $(h, m)$, we marginalize over $\varphi$ and $\tau$ for a particular choice of $y$:

$$p(y | h, m, \mathcal{D}) = \iint p(y, \tau, \varphi | h, m, \mathcal{D}) d\varphi d\tau \tag{9}$$

$$= \iint p(y | \tau, \varphi, h, m, \mathcal{D}) p(\tau | \mathcal{D}) p(\varphi | \mathcal{D}) d\varphi d\tau \tag{10}$$

The second line is obtained by conditioning on $\tau, \varphi$ and using the fact that $\tau$ and $\varphi$ are independent given $\mathcal{D}$. We now use Equation (2) to re-write the first term, obtaining (up to a constant):

$$\propto \iint p(h | y, \varphi) p(y | m, \tau) p(\tau | \mathcal{D}) p(\varphi | \mathcal{D}) d\varphi d\tau \tag{11}$$

We now split the integral into its independent components, and use our parametric assumptions to replace $p(h | y, \varphi)$ with $\varphi_{hy}$ and $p(y | m, \tau)$ with $m_y^{(\tau)}$:

$$= \left[ \int m_y^{(\tau)} p(\tau | \mathcal{D}) d\tau \right] \left[ \int \varphi_{hy} p(\varphi | \mathcal{D}) d\varphi \right] \tag{12}$$

The second integral is the posterior mean of $\varphi_{hy}$, which is available in closed-form by conjugacy. However, as we do not have a closed-form posterior for $p(\tau | \mathcal{D})$, we estimate the first integral using our samples. In all, we obtain

$$\approx \left[ \frac{1}{n_s} \sum_{j=1}^{n_s} m_y^{(\tau_j)} \right] \cdot \frac{\alpha'_{hy}}{\sum_{\ell=1}^{K} \alpha'_{\ell j}} \tag{13}$$

where $\alpha'_{ij}$ is the posterior Dirichlet parameter for entry $(i, j)$ in the confusion matrix $\varphi$. Note that the resulting probabilities will be un-normalized, but normalization is straightforward as we are considering a set of discrete outcomes.

In practice, we use HMC [Neal et al., 2011] in the Pyro probabilistic programming language [Bingham et al., 2019] to sample from the posterior over log-temperatures.

## Appendix C   EM Algorithm Details

In this section, we provide a detailed derivation and description of our EM algorithm.

Let $\mathcal{D}_C = \{(m(x_\ell), h(x_\ell))\}_{\ell=1}^n$ be an unlabeled dataset used for fitting combination parameters, consisting of classifier probabilities and human labels but no ground truth labels. Our goal is to infer classifier calibration parameters $\theta$ and the human confusion matrix $\varphi$ from $\mathcal{D}_C$. We use $m_\ell$ as a shorthand for $m(x_\ell)$ throughout (respectively for $h$).

We can fit this model via EM, where the ground truth is treated as latent. For simplicity, we derive the maximum likelihood variant, and discuss the necessary changes for the MAP variant at the end of this section. In the E-step, $p(y|m_\ell, h_\ell, \varphi, \theta)$ is estimated from Equation (3).

For the M-step, we maximize the expected log-likelihood, where we use $\Theta = \{\theta, \varphi\}$ to denote the set of all parameters:

$$
\begin{aligned}
\Theta_{t+1} &= \arg\max_\Theta \sum_i \mathbb{E}_{y \sim p(y|h,m,\theta_t)} \left[ \log p(y, h_\ell, m_\ell | \Theta) \right] \\
&= \arg\max_\Theta \sum_\ell \sum_y p(y|h_\ell, m_\ell, \Theta_t) \log p(y, h_\ell, m_\ell | \Theta) \\
&= \arg\max_\Theta \left[ \sum_\ell \sum_y p(y|h_\ell, m_\ell, \Theta_t) \log p(h_\ell | y, \Theta) \right. \\
&\qquad\qquad \left. + \sum_\ell \sum_y p(y|h_\ell, m_\ell, \Theta_t) \log p(y|m_\ell, \Theta) + C \right]
\end{aligned}
$$

where $C$ is a constant not depending on $\Theta$. Assuming further that the calibration and confusion parameters are independent, the M-step becomes two independent optimizations (i.e. one for $\theta$ and one for $\varphi$):

$$
\theta_{t+1} = \arg\max_\theta \sum_\ell \sum_y p(y|h_\ell, m_\ell, \Theta_t) \log p(y|m_\ell, \theta) \tag{14}
$$

$$
\varphi_{t+1} = \arg\max_\varphi \sum_\ell \sum_y p(y|h_\ell, m_\ell, \Theta_t) \log p(h_\ell | y, \varphi) \tag{15}
$$

In Equation (14), $\log p(y|m_\ell, \theta)$ depends on the calibration method we choose, and the update for $\theta_{t+1}$ does not have a closed-form update. We use gradient methods to maximize this term.

Equation (15) is maximum likelihood for the confusion matrix and hence $\varphi_{t+1}$ can be solved for in closed-form. In particular, the value for $\varphi_{t+1}$ at entry $i, j$ is

$$
\varphi_{i,j} = \frac{\sum_{\ell: h_\ell = a} p(y = j | h_\ell, m_\ell, \Theta_t)}{\sum_\ell p(y = j | h_\ell, m_\ell, \Theta_t)} \tag{16}
$$

For the MAP variant of our EM algorithm, our optimizations become

$$
\theta_{t+1} = \arg\max_\theta \sum_\ell \sum_y p(y|h_\ell, m_\ell, \Theta_t) \log p(y|m_\ell, \theta) + \log p(\theta) \tag{17}
$$

$$
\varphi_{t+1} = \arg\max_\varphi \sum_\ell \sum_y p(y|h_\ell, m_\ell, \Theta_t) \log p(h_\ell | y, \varphi) + \log p(\varphi) \tag{18}
$$

The first optimization (Equation (17)) is still fit using gradient methods. As we choose independent Dirichlet priors for each column of $\varphi$, the closed-form estimate for $\varphi$ becomes

$$
\varphi_{i,j} = \frac{\alpha_{ji} - 1 + \sum_{\ell: h_\ell = a} p(y = j | h_\ell, m_\ell, \Theta_t)}{\gamma + (K - 1)\beta - K + \sum_\ell p(y = j | h_\ell, m_\ell, \Theta_t)} \tag{19}
$$

which is analogous to the typical Dirichlet-multinomial posterior.

## Appendix D    Conditional Independence Combination as a Special Case of Logistic Regression

We demonstrate that the conditional independence combination (Equation (3)) can be seen as a special case of logistic regression taking $m(x)$ and $h(x)$ as inputs, but only when the calibration map takes a particular functional form. Calibration maps such as temperature scaling and Dirichlet calibration [Kull et al., 2019] satisfy this requirement.

### D.1    Logistic Regression

In the logistic regression (LR) model, for input $x$ we have features $z \in \mathbb{R}^{2k}$, $z(x) = m(x) \oplus H(x)$, where $H(x)$ is the one-hot version of $h(x)$ and $\oplus$ is the direct sum. A weight matrix $W \in \mathbb{R}^{k \times 2k}$ and a bias $b \in \mathbb{R}^k$ are to be learned. The probabilistic output is given by an element-wise softmax:

$$x \mapsto \text{SoftMax}(Wz(x) + b) \in \mathbb{R}^k \tag{20}$$

We can write $W = [W_m | W_h]$ as a block matrix, where $W_m, W_h \in \mathbb{R}^{k \times k}$ are the model and human weights respectively. In log-space, the LR model is then

$$\log p(y | m(x), h(x)) = W_m m(x) + W_h H(x) + b - \log(C) \tag{21}$$

where $C$ is a normalizing constant. Since $H(x)$ is one-hot, the term $W_h H(x)$ corresponds to a column in $W_h$, e.g. if $H(x) = [1, 0, \ldots, 0]^\mathsf{T}$, then $W_h H(x)$ is the first column of $W_h$. The above is the full vector of probabilities. To make it clearer, for an index $i$, let $W_m^i$ be the $i$th row of $W_m$ (resp. for $W_h$).

$$p(y = i | m(x), h(x)) = W_m^i m(x) + W_h^i H(x) + b_i - \log(C) \tag{22}$$
$$= W_m^i m(x) + (W_h)_{ih(x)} + b_i - \log(C) \tag{23}$$

### D.2    CI Model

In the CI model,
$$p(y | m(x), h(x)) \propto p(y | m(x)) p(h(x) | y) \tag{24}$$
In log-space for a single index $i$:

$$\log p(y = i | m(x), h(x)) = \log p(y | m(x)) + \log p(h(x) | y) - \log(C) \tag{25}$$
$$= \log m_i^\theta(x) + \log \varphi_{h(x)y} - \log(C) \tag{26}$$

### D.3

From this, we see that $W_h$ is analogous to the log-confusion matrix of $h$. Similarly, $W_m$ can be thought of as a linear operator mapping the model probabilities to log-calibrated model probabilities.

If we use $\log m(x)$ (pointwise) for the input feature $z(x)$, the LR model is

$$p(y = i | m(x), h(x)) = W_m^i \log m(x) + (W_h)_{ih(x)} + b_i - \log(C) \tag{27}$$

In the special case $W_m = \frac{1}{T} I$, $b_i = 0$, and $W_h = \log \varphi^\mathsf{T}$, we recover temperature scaling CI. In fact, the equation $W_m \log m(x) + b$ is the same as Dirichlet calibration – vector scaling / matrix scaling are special cases as well.

## Appendix E  Learning Curves

In addition to those in Figure 2, we provide learning curves that include additional baseline models: logistic regression (LR), the single-parameter confusion matrix method (SP), and the fully Bayesian P+L method (P+L Fully Bayesian). We report only the mean error rate averaged over 10 random seeds for the sake of visual clarity. All methods (other than LR) are fit using MAP inference. We do not present the maximum likelihood (ML) variants for these methods, as the MAP methods outperform their ML counterparts in our experiments.

While the SP method is label efficient given its low parameter count, it often underfits to the data and converges to an error rate worse than the P+L method. In some cases, the fully Bayesian obtains a lower error rate than the P+L method, but requires more labeled data to be fit, as well as being more computationally intensive. On the CIFAR-10 data, the logistic regression method is label inefficient, and while it outperforms the L+L method, converges to a worse error rate than the P+L method. In contrast, LR is able to outperform the P+L method on the ImageNet-16H datasets, but only when fit several hundred datapoints.

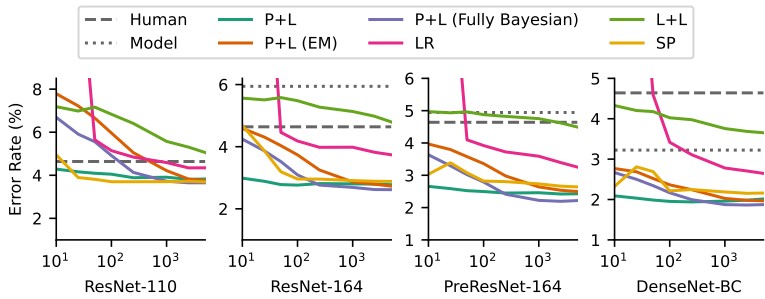

Figure 7: Learning curves for various models on CIFAR-10H.

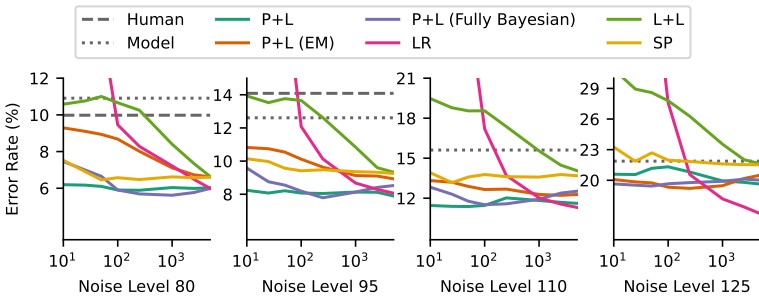

Figure 8: Learning curves for VGG-19 on ImageNet-16H at various noise levels.

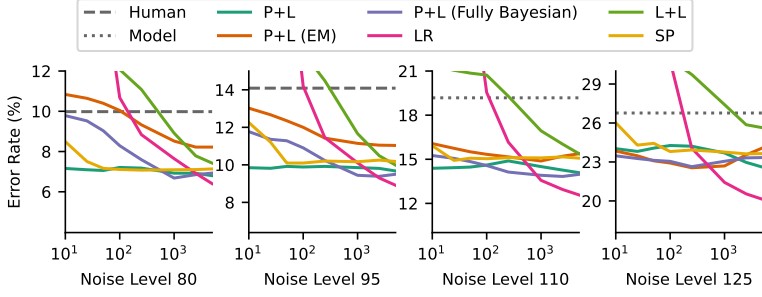

Figure 9: Learning curves for GoogLeNet on ImageNet-16H at various noise levels.

# Appendix F  Dataset, Model Training, and Code Details

## F.1  CIFAR-10H

The CIFAR-10H dataset Peterson et al. [2019] consists of the $10,000$ images in the standard CIFAR-10 test set, but each image is labeled by approximately $50$ individual human labelers. There are ten classes in this dataset.

We study four CNN model architectures on CIFAR-10H:

- ResNet-110 and ResNet-164 He et al. [2016a]: Deep residual networks with 110 and 164 layers respectively.
- PreResNet-164 He et al. [2016b]: A deep residual network with identity mappings as skip connections, with 164 layers.
- DenseNet-BC Huang et al. [2017]: A densely connected CNN with $L = 190$ layers and a growth-rate of $k = 40$, using bottleneck layers.

For each model, we use pre-trained weights available at `https://github.com/bearpaw/pytorch-classification` (MIT License). These models were trained on the standard CIFAR-10 training split.

## F.2  ImageNet-16H

The ImageNet-16H dataset consists of noisy images from the ImageNet test set Deng et al. [2009], distorted by phase noise at each spatial frequency based on four levels of phase noise (80, 95, 110, and 125). Approximately 7200 images were classified at each noise level (with slight variability per noise level). The number of classes is reduced to 16 (as compared to 1000 in the original ImageNet dataset).

We study two model architectures on ImageNet-16H: VGG-19 Simonyan and Zisserman [2015] and GoogLeNet Szegedy et al. [2015]. Our training procedure is detailed as follows. We first load a pre-trained ImageNet model (trained on the original 1000 class ImageNet dataset) from the PyTorch model library [Paszke et al., 2019]. We remove the final linear layer and replace it with a randomly initialized linear layer with a 16-dimensional output. We then fine-tune all model weights (using the cross-entropy loss) on noisy images from the ImageNet-16H training set (261,168 images). The models are fine-tuned to all levels of noise simultaneously by randomly assigning a different degree of phase noise (ranging from 0 to 130 degrees) to each training image in a batch.

## F.3  Additional Code Details

Our experiments are implemented in Python 3.8, and make use of the following libraries:

- Scikit-Learn [Pedregosa et al., 2011] (BSD License)
- PyTorch [Paszke et al., 2019] (BSD License)
- Pyro [Bingham et al., 2019] (Apache 2.0 License)
- NumPy [Harris et al., 2020] (BSD License)
- IMax Calibration Patel et al. [2021] (AGPL-3.0 License)
- Ensemble Temperature Scaling Zhang et al. [2020] (MIT License)

## F.4  Compute Resources

All of our experiments were conducted on a standard desktop computer (AMD Ryzen 5 6-core @ 3.6GHz, 16GB memory).

Other than the fully Bayesian combination, all combination methods studied in this work do not require significant computational resources and can be fit on the order of seconds. The fully Bayesian method (Appendix B) is more computationally intensive as it requires the use of MCMC to sample from the posterior distribution over calibration parameters, but can still be fit in approx. 2 minutes with 5000 labeled datapoints. However, we focus on MAP estimation in our main results (which

does not require MCMC), and only compare to the fully Bayesian setup as a baseline comparison. In addition, we find the fully Bayesian setup to be less label efficient than the MAP counterpart (see Appendix E).

In terms of model training, our ImageNet-16H models were trained on an internal GPU server with 8x GTX 2080ti GPUs and 2 x Intel Xeon Gold 5218 (16 core) processors. On our hardware, fine-tuning for 50 epochs requires approximately 6 hours of training per model.

## Appendix G   Individual-Level Combinations

Here, we investigate the robustness of our method to the sampling used to select human-generated labels for the images in our experiments. In particular, we fit a confusion matrix for each individual human annotator on both the CIFAR-10H and ImageNet-16H datasets.

The combination is fit with 25 ground-truth labeled datapoints, and is evaluated using 25 datapoints on CIFAR-10H and 175 images on ImageNet-16H. The combination method is our Bayesian P+L combination. The train and test sizes are small here, as each individual human only labels a few images in our datasets. The reported results are averaged across all human labelers for ten random train/test splits.

In general, we see that the individual-level combinations obtain a similar level of performance to that of the sampled combinations (i.e. compared to the results in Appendix H).

| Model Name | Human | Model | P+L |
|---|---|---|---|
| ResNet-110 | $5.11 \pm 5.7$ | $11.1 \pm 2.5$ | $4.06 \pm 1.76$ |
| ResNet-164 | — | $6.09 \pm 1.75$ | $2.87 \pm 1.4$ |
| PreResNet-164 | — | $4.92 \pm 1.6$ | $2.62 \pm 1.3$ |
| DenseNet-BC | — | $3.31 \pm 1.4$ | $2.04 \pm 1.1$ |

Table 4: Error rates on CIFAR-10H, $\pm$ one standard deviation. We fit a confusion matrix to each individual labeler using 25 datapoints, and evaluated with the remaining data (175 points per individual). The combination is the MAP P+L method, using the priors detailed in the main body of the paper.

| Noise Level | Human | Model | P+L |
|---|---|---|---|
| 80 | $9.81 \pm 8.5$ | $11.29 \pm 6.3$ | $6.15 \pm 4.9$ |
| 95 | $13.87 \pm 9.7$ | $12.56 \pm 6.3$ | $8.11 \pm 5.5$ |
| 110 | $23.13 \pm 12.3$ | $15.49 \pm 7.6$ | $11.59 \pm 6.7$ |
| 125 | $39.84 \pm 13.4$ | $21.92 \pm 8.2$ | $20.87 \pm 8.3$ |

Table 5: Error rates with VGG19 on ImageNet-16H, $\pm$ one standard deviation. We fit a confusion matrix to each individual labeler using 25 datapoints, and evaluated with the remaining data (25 points per individual). The combination is the MAP P+L method, using the priors detailed in the main body of the paper.

| Noise Level | Human | Model | P+L |
|---|---|---|---|
| 80 | $9.81 \pm 8.5$ | $14.49 \pm 7.0$ | $7.11 \pm 5.6$ |
| 95 | $13.87 \pm 9.7$ | $17.16 \pm 7.0$ | $9.72 \pm 6.4$ |
| 110 | $23.13 \pm 12.3$ | $18.88 \pm 8.2$ | $14.25 \pm 7.7$ |
| 125 | $39.84 \pm 13.4$ | $26.63 \pm 9.2$ | $24.30 \pm 9.3$ |

Table 6: Error rates with DenseNet-BC on ImageNet-16H, $\pm$ one standard deviation. We fit a confusion matrix to each individual labeler using 25 datapoints, and evaluated with the remaining data (25 points per individual). The combination is the MAP P+L method, using the priors detailed in the main body of the paper.

# Appendix H   Calibration Methods and Uncalibrated Combinations

In this section we provide additional empirical results on CIFAR-10H and ImageNet-16H. In particular, we evaluate several different calibration methods (MAP TS (as used in the main paper) Guo et al. [2017], Ensemble TS Zhang et al. [2020], IMax Binning Patel et al. [2021]). We also compare to the L+L combination, and the P+L combination of the uncalibrated model probabilities with the human labels (Uncalibrated). The error rate of the human alone (Human) and model alone (Model) are provided for context.

In most cases, human-machine combinations using calibrated probabilities outperform those using uncalibrated probabilities. Moreover, in some cases we obtain small gains in performance by using a more complex calibration map (IMax Binning), but it is not clear how to incorporate prior information with this method. As prior information is useful in increasing the label efficiency and decreasing the error rate of the combination, our focus in the main paper is on MAP TS as our calibration method.

All tables in this section correspond to error rates ($\pm$ one standard deviation) averaged across 25 different random seeds. The combinations (P+L, Equation (3)) are fit using 5000 labeled data points on CIFAR-10H and using between 5067 and 5152 data points on ImageNet-16H (varies by noise level). The combinations are evaluated using 3000 data points on CIFAR-10H and using between 2171 and 2208 on ImageNet-16H.

| Model Name | Human | Model | Combination | | | | |
| --- | --- | --- | --- | --- | --- | --- | --- |
| | | | L+L | Uncalibrated | TS | ETS | IMax |
| ResNet-110 | $4.62 \pm 0.33$ | $11.28 \pm 0.44$ | $4.70 \pm 0.36$ | $4.40 \pm 0.25$ | $3.83 \pm 0.15$ | $3.76 \pm 0.25$ | $\mathbf{3.80 \pm 0.24}$ |
| ResNet-164 | — | $6.10 \pm 0.38$ | $4.71 \pm 0.37$ | $3.05 \pm 0.23$ | $\mathbf{2.78 \pm 0.15}$ | $2.82 \pm 0.23$ | $2.85 \pm 0.23$ |
| PreResNet-164 | — | $5.00 \pm 0.36$ | $4.36 \pm 0.39$ | $2.90 \pm 0.22$ | $\mathbf{2.43 \pm 0.22}$ | $2.46 \pm 0.25$ | $\mathbf{2.43 \pm 0.26}$ |
| DenseNet-BC | — | $3.25 \pm 0.30$ | $3.39 \pm 0.32$ | $2.22 \pm 0.21$ | $\mathbf{2.01 \pm 0.15}$ | $2.17 \pm 0.17$ | $2.04 \pm 0.18$ |

Table 7: Error rates (%, $\pm$ one standard deviation) averaged over 25 seeds on CIFAR-10H for various classifiers.

| Human | Model | Combination | | | | |
| --- | --- | --- | --- | --- | --- | --- |
| | | L+L | Uncalibrated | TS | ETS | IMax |
| $9.99 \pm 0.48$ | $11.10 \pm 0.60$ | $6.78 \pm 0.42$ | $7.52 \pm 0.52$ | $\mathbf{6.03 \pm 0.54}$ | $6.79 \pm 0.44$ | $6.31 \pm 0.46$ |
| $14.07 \pm 0.70$ | $12.58 \pm 0.53$ | $9.01 \pm 0.57$ | $9.02 \pm 0.44$ | $\mathbf{7.89 \pm 0.37}$ | $9.32 \pm 0.49$ | $8.62 \pm 0.47$ |
| $22.99 \pm 0.71$ | $15.51 \pm 0.62$ | $14.07 \pm 0.82$ | $12.59 \pm 0.53$ | $\mathbf{11.62 \pm 0.54}$ | $13.18 \pm 0.59$ | $12.30 \pm 0.57$ |
| $39.76 \pm 0.75$ | $22.07 \pm 0.69$ | $21.89 \pm 0.66$ | $\mathbf{19.45 \pm 0.62}$ | $19.63 \pm 0.70$ | $20.74 \pm 0.62$ | $20.47 \pm 0.63$ |

Table 8: Error rates (%, $\pm$ one standard deviation) averaged over 25 seeds, VGG-19 on ImageNet-16H. Each row corresponds to a different noise level (80, 95, 110, 125).

| Human | Model | Combination | | | | |
| --- | --- | --- | --- | --- | --- | --- |
| | | L+L | Uncalibrated | TS | ETS | IMax |
| $9.99 \pm 0.48$ | $14.48 \pm 0.70$ | $7.33 \pm 0.39$ | $8.06 \pm 0.50$ | $\mathbf{6.80 \pm 0.47}$ | $7.73 \pm 0.41$ | $7.66 \pm 0.44$ |
| $14.07 \pm 0.70$ | $17.22 \pm 0.72$ | $9.93 \pm 0.73$ | $10.66 \pm 0.52$ | $\mathbf{9.67 \pm 0.40}$ | $10.23 \pm 0.51$ | $10.05 \pm 0.49$ |
| $22.99 \pm 0.71$ | $19.09 \pm 0.75$ | $15.43 \pm 0.71$ | $14.78 \pm 0.64$ | $\mathbf{14.09 \pm 0.55}$ | $14.76 \pm 0.53$ | $14.53 \pm 0.53$ |
| $39.76 \pm 0.75$ | $27.06 \pm 0.47$ | $25.64 \pm 0.43$ | $23.06 \pm 0.72$ | $\mathbf{22.60 \pm 0.63}$ | $24.38 \pm 0.64$ | $23.91 \pm 0.69$ |

Table 9: Error rates (%, $\pm$ one standard deviation) averaged over 25 seeds, GoogLeNet on ImageNet-16H. Each row corresponds to a different noise level (80, 95, 110, 125).

# Appendix I    Calibration Properties of Combinations

We further study the calibration properties of human-machine (P+L) combinations. The results in this Appendix are analogous to the results in Table 2 for our ImageNet-16H models, where we show various calibration metrics as we vary the number of labeled datapoints used for fitting the combination. In general, we find that using only a small number of labeled datapoints (10 in our experiments) is sufficient, and we do not observe further improvements in calibration by using more labeled data (5000 points in our experiments) to fit the combination.

In addition, we investigate whether the resulting human-machine combination can be further calibrated. We calibrate the resulting human-machine combinations (with MAP TS) using the same data used to fit the combination, i.e. 5000 labeled datapoints (Recal. Comb.). We find that it is possible to further reduce the ECE of the combinations, but other metrics only see small improvements. However, we note that this does not affect the error rate of the combination, as MAP TS is accuracy-preserving.

| Metric | Model Name | No Calibration | | 10 Datapoints | | 5000 Datapoints | | |
| | | Model | Comb. | Model | Comb. | Model | Comb. | Recal. Comb. |
|---|---|---|---|---|---|---|---|---|
| ECE ($10^{-2}$) | ResNet-110 | $5.23 \pm 0.35$ | $2.08 \pm 0.25$ | $3.03 \pm 0.58$ | $1.30 \pm 0.23$ | $2.99 \pm 0.36$ | $1.76 \pm 0.18$ | $0.85 \pm 0.22$ |
| | ResNet-164 | $2.98 \pm 0.34$ | $1.63 \pm 0.23$ | $1.95 \pm 0.33$ | $1.25 \pm 0.18$ | $1.89 \pm 0.32$ | $1.39 \pm 0.18$ | $0.84 \pm 0.20$ |
| | PreResNet-164 | $3.03 \pm 0.29$ | $1.87 \pm 0.22$ | $2.31 \pm 0.33$ | $1.40 \pm 0.26$ | $2.27 \pm 0.31$ | $1.43 \pm 0.21$ | $1.06 \pm 0.21$ |
| | DenseNet-BC | $2.18 \pm 0.27$ | $1.53 \pm 0.20$ | $1.76 \pm 0.28$ | $1.34 \pm 0.14$ | $1.73 \pm 0.28$ | $1.27 \pm 0.13$ | $0.95 \pm 0.18$ |
| cwECE ($10^{-2}$) | ResNet-110 | $0.81 \pm 0.07$ | $0.23 \pm 0.05$ | $0.58 \pm 0.07$ | $0.24 \pm 0.05$ | $0.58 \pm 0.06$ | $0.19 \pm 0.06$ | $0.19 \pm 0.04$ |
| | ResNet-164 | $0.39 \pm 0.06$ | $0.15 \pm 0.03$ | $0.31 \pm 0.05$ | $0.15 \pm 0.04$ | $0.31 \pm 0.05$ | $0.13 \pm 0.03$ | $0.14 \pm 0.03$ |
| | PreResNet-164 | $0.29 \pm 0.04$ | $0.13 \pm 0.03$ | $0.28 \pm 0.04$ | $0.13 \pm 0.03$ | $0.28 \pm 0.04$ | $0.13 \pm 0.03$ | $0.13 \pm 0.03$ |
| | DenseNet-BC | $0.23 \pm 0.03$ | $0.11 \pm 0.02$ | $0.24 \pm 0.02$ | $0.12 \pm 0.02$ | $0.24 \pm 0.02$ | $0.11 \pm 0.02$ | $0.10 \pm 0.02$ |
| NLL | ResNet-110 | $0.40 \pm 0.02$ | $0.16 \pm 0.01$ | $0.35 \pm 0.02$ | $0.15 \pm 0.01$ | $0.35 \pm 0.02$ | $0.14 \pm 0.01$ | $0.12 \pm 0.01$ |
| | ResNet-164 | $0.24 \pm 0.02$ | $0.11 \pm 0.01$ | $0.20 \pm 0.01$ | $0.10 \pm 0.01$ | $0.20 \pm 0.01$ | $0.10 \pm 0.01$ | $0.09 \pm 0.01$ |
| | PreResNet-164 | $0.23 \pm 0.02$ | $0.13 \pm 0.02$ | $0.19 \pm 0.02$ | $0.11 \pm 0.01$ | $0.19 \pm 0.02$ | $0.10 \pm 0.01$ | $0.08 \pm 0.01$ |
| | DenseNet-BC | $0.17 \pm 0.01$ | $0.10 \pm 0.01$ | $0.14 \pm 0.01$ | $0.09 \pm 0.01$ | $0.14 \pm 0.01$ | $0.08 \pm 0.01$ | $0.07 \pm 0.01$ |

Table 10: Calibration metrics on CIFAR-10H.

| Metric | Noise Level | No Calibration | | 10 Datapoints | | 5000 Datapoints | | |
| | | Model | Comb. | Model | Comb. | Model | Comb. | Recal. Comb |
|---|---|---|---|---|---|---|---|---|
| ECE ($10^{-2}$) | 80 | $8.54 \pm 0.54$ | $5.17 \pm 0.49$ | $7.30 \pm 0.69$ | $3.91 \pm 0.53$ | $7.15 \pm 0.60$ | $4.01 \pm 0.42$ | $3.17 \pm 0.42$ |
| | 95 | $8.96 \pm 0.48$ | $5.72 \pm 0.39$ | $7.49 \pm 0.77$ | $4.93 \pm 0.36$ | $7.26 \pm 0.51$ | $4.56 \pm 0.37$ | $3.23 \pm 0.35$ |
| | 110 | $9.76 \pm 0.53$ | $7.81 \pm 0.48$ | $7.81 \pm 0.91$ | $6.31 \pm 0.67$ | $7.24 \pm 0.56$ | $6.07 \pm 0.53$ | $3.92 \pm 0.49$ |
| | 125 | $11.81 \pm 0.64$ | $10.89 \pm 0.52$ | $7.34 \pm 1.45$ | $10.21 \pm 0.64$ | $7.29 \pm 0.56$ | $8.46 \pm 0.68$ | $4.49 \pm 0.60$ |
| cwECE ($10^{-2}$) | 80 | $1.10 \pm 0.07$ | $0.68 \pm 0.05$ | $1.01 \pm 0.07$ | $0.59 \pm 0.06$ | $1.01 \pm 0.06$ | $0.54 \pm 0.05$ | $0.56 \pm 0.04$ |
| | 95 | $1.18 \pm 0.06$ | $0.82 \pm 0.05$ | $1.13 \pm 0.06$ | $0.73 \pm 0.06$ | $1.12 \pm 0.06$ | $0.72 \pm 0.04$ | $0.69 \pm 0.04$ |
| | 110 | $1.44 \pm 0.06$ | $1.14 \pm 0.06$ | $1.38 \pm 0.07$ | $1.04 \pm 0.08$ | $1.36 \pm 0.07$ | $1.03 \pm 0.07$ | $0.96 \pm 0.05$ |
| | 125 | $1.98 \pm 0.06$ | $1.73 \pm 0.07$ | $1.86 \pm 0.05$ | $1.54 \pm 0.07$ | $1.85 \pm 0.04$ | $1.52 \pm 0.06$ | $1.45 \pm 0.06$ |
| NLL | 80 | $0.71 \pm 0.05$ | $0.49 \pm 0.04$ | $0.53 \pm 0.05$ | $0.37 \pm 0.04$ | $0.52 \pm 0.03$ | $0.34 \pm 0.03$ | $0.27 \pm 0.02$ |
| | 95 | $0.70 \pm 0.03$ | $0.52 \pm 0.03$ | $0.55 \pm 0.04$ | $0.41 \pm 0.03$ | $0.54 \pm 0.03$ | $0.39 \pm 0.02$ | $0.32 \pm 0.02$ |
| | 110 | $0.73 \pm 0.03$ | $0.61 \pm 0.04$ | $0.60 \pm 0.04$ | $0.51 \pm 0.05$ | $0.57 \pm 0.03$ | $0.49 \pm 0.04$ | $0.41 \pm 0.02$ |
| | 125 | $0.89 \pm 0.03$ | $0.83 \pm 0.04$ | $0.75 \pm 0.03$ | $0.77 \pm 0.03$ | $0.74 \pm 0.02$ | $0.71 \pm 0.03$ | $0.64 \pm 0.02$ |

Table 11: Calibration metrics for VGG-19 on ImageNet-16H.

| Metric | Noise Level | No Calibration | | 10 Datapoints | | 5000 Datapoints | | |
| | | Model | Comb. | Model | Comb. | Model | Comb. | Recal. Comb. |
|---|---|---|---|---|---|---|---|---|
| ECE ($10^{-2}$) | 80 | $7.39 \pm 0.58$ | $4.10 \pm 0.47$ | $4.60 \pm 0.62$ | $3.04 \pm 0.32$ | $4.52 \pm 0.58$ | $3.07 \pm 0.40$ | $1.97 \pm 0.39$ |
| | 95 | $9.23 \pm 0.58$ | $5.68 \pm 0.40$ | $5.91 \pm 0.55$ | $4.19 \pm 0.54$ | $5.76 \pm 0.39$ | $4.32 \pm 0.42$ | $2.51 \pm 0.45$ |
| | 110 | $9.04 \pm 0.68$ | $7.60 \pm 0.46$ | $5.34 \pm 1.15$ | $6.66 \pm 0.71$ | $5.34 \pm 0.37$ | $5.98 \pm 0.50$ | $3.00 \pm 0.43$ |
| | 125 | $11.98 \pm 0.40$ | $10.95 \pm 0.62$ | $6.78 \pm 1.40$ | $11.33 \pm 0.36$ | $6.54 \pm 0.43$ | $7.98 \pm 0.60$ | $3.37 \pm 0.42$ |
| cwECE ($10^{-2}$) | 80 | $1.33 \pm 0.07$ | $0.67 \pm 0.04$ | $1.30 \pm 0.07$ | $0.63 \pm 0.04$ | $1.30 \pm 0.07$ | $0.57 \pm 0.03$ | $0.57 \pm 0.03$ |
| | 95 | $1.47 \pm 0.08$ | $0.87 \pm 0.05$ | $1.46 \pm 0.05$ | $0.78 \pm 0.06$ | $1.47 \pm 0.05$ | $0.75 \pm 0.04$ | $0.74 \pm 0.05$ |
| | 110 | $1.70 \pm 0.08$ | $1.23 \pm 0.06$ | $1.66 \pm 0.08$ | $1.12 \pm 0.07$ | $1.65 \pm 0.04$ | $1.10 \pm 0.06$ | $1.03 \pm 0.06$ |
| | 125 | $2.31 \pm 0.06$ | $1.92 \pm 0.06$ | $2.19 \pm 0.06$ | $1.70 \pm 0.06$ | $2.19 \pm 0.06$ | $1.68 \pm 0.05$ | $1.59 \pm 0.06$ |
| NLL | 80 | $0.59 \pm 0.03$ | $0.34 \pm 0.02$ | $0.50 \pm 0.02$ | $0.29 \pm 0.02$ | $0.50 \pm 0.03$ | $0.28 \pm 0.02$ | $0.25 \pm 0.02$ |
| | 95 | $0.65 \pm 0.03$ | $0.43 \pm 0.03$ | $0.56 \pm 0.01$ | $0.37 \pm 0.02$ | $0.56 \pm 0.01$ | $0.36 \pm 0.02$ | $0.33 \pm 0.02$ |
| | 110 | $0.75 \pm 0.03$ | $0.60 \pm 0.03$ | $0.66 \pm 0.03$ | $0.56 \pm 0.03$ | $0.66 \pm 0.02$ | $0.53 \pm 0.02$ | $0.47 \pm 0.02$ |
| | 125 | $0.97 \pm 0.02$ | $0.88 \pm 0.03$ | $0.85 \pm 0.02$ | $0.89 \pm 0.02$ | $0.85 \pm 0.01$ | $0.79 \pm 0.02$ | $0.74 \pm 0.02$ |

Table 12: Calibration metrics for GoogLeNet on ImageNet-16H.

# Appendix J  Proofs of Theorem (1) and Theorem (2)

We provide proofs for our theoretical claims in Section 6.

## J.1  Confidence Ratios

*Proof of Theorem* (1). Recall that $c(x)$ is the prediction output by Equation (3). The accuracy is then bounded as follows:

$$
\begin{aligned}
\mathbb{E}\left[\mathbb{1}\left(c(x)=y\right)\right] &= \mathbb{P}\left\{y=\arg\max_{k}\varphi_{h(x)k}m_{k}^{\theta}(x)\right\} \\
&= \mathbb{P}\left\{\varphi_{h(x)y}m_{y}^{\theta}(x)>\max_{k\neq y}\varphi_{h(x)k}m_{k}^{\theta}(x)\right\} \\
&\geq \mathbb{P}\left\{\varphi_{h(x)y}m_{y}^{\theta}(x)>\max_{k\neq y}\varphi_{h(x)k}\max_{k\neq y}m_{k}^{\theta}(x)\right\} \\
&\geq \mathbb{P}\left\{\varphi_{h(x)y}m_{y}^{\theta}(x)>\left(1-\varphi_{h(x)y}\right)\left(1-m_{y}^{\theta}(x)\right)\right\} \\
&= \mathbb{P}\left\{r_{m}(x)>\left(r_{h}(x)\right)^{-1}\right\}
\end{aligned}
$$

In fact, we have proved the stronger but somewhat less interpretable inequality:

$$
\mathbb{E}\left[\mathbb{1}\left(c(x)=y\right)\right]\geq\mathbb{P}\left\{\frac{m_{y}^{\theta}(x)}{\max_{k\neq y}m_{k}^{\theta}(x)}>\left(\frac{\varphi_{h(x)y}}{\max_{k\neq y}\varphi_{h(x)k}}\right)^{-1}\right\}
$$

$\square$

We note further that the same argument can be used to analyze the combination of two probabilistic predictors when the combination is done by pointwise multiplying their calibrated probabilities. In particular, if we have two probabilistic classifiers $m$ and $\widetilde{m}$,

$$
\mathbb{E}\left[\mathbb{1}\left(c(x)=y\right)\right]\geq\mathbb{P}\left\{r_{m}(x)>\left(r_{\widetilde{m}}(x)\right)^{-1}\right\}
$$

where $r_{\widetilde{m}}$ is defined analogously to $r_{m}$. The proof for this statement is exactly analogous to that of Theorem (1), where $\widetilde{m}_{y}^{\widetilde{\theta}}$ now plays the role of $\varphi_{h(x)y}$. This same argument can again be adapted for the combination of two non-probabilistic combiners, combined by parameterizing Equation (2) with their confusion matrices.

## J.2  Estimation Error

We begin with a useful lemma that will play a key part in our estimation error analysis.

**Lemma 1.** *For scalars $a_1, a_2, b_1, b_2 \in [0,1]$, the difference of the products is at most the sum of the differences:*

$$
|a_1 b_1 - a_2 b_2| \leq |a_1 - a_2| + |b_1 - b_2| \tag{28}
$$

*Proof.*

$$
\begin{aligned}
|a_1 b_1 - a_2 b_2| &= |a_1 b_1 - a_2 b_2 + a_1 b_2 - a_1 b_2| \\
&= |a_1(b_1 - b_2) + b_2(a_1 - a_2)| \\
&\leq |a_1|\cdot|b_1 - b_2| + |b_2|\cdot|a_1 - a_2| \quad\text{(triangle inequality)} \\
&\leq |b_1 - b_2| + |a_1 - a_2|
\end{aligned}
$$

$\square$

We now proceed to the proof of Theorem (2).

*Proof of Theorem* (2).  Recall that $\eta(x, y) = |p(h(x)|y)p(y|m(x)) - m_y^\theta(x)\widehat{\varphi}_{h(x)y}|$ is the estimation error for Equation (3) (up to normalizing constants), where $\widehat{\varphi}_{ij}$ represents an estimate of $p(h(x) = i|y = j)$.

By the law of total expectation, we can condition on a particular value of $y$ and $h(x)$:

$$\mathbb{E}\left[\eta(x, y)\right] = \sum_{i=1}^{K}\sum_{j=1}^{K} p(y = j)\varphi_{ij}\mathbb{E}\left[\eta(x, y)|y = j, h(x) = i\right] \tag{29}$$

We now apply Lemma (1) to the conditional expectation above:

$$\mathbb{E}\left[\eta(x, y)|y = j, h(x) = i\right]$$
$$= \mathbb{E}\left[\left.\left|\varphi_{ij}p(y = j|m(x)) - \widehat{\varphi}_{ij}m_j^\theta(X)\right|\right|y = j, h(x) = i\right]$$
$$\leq \mathbb{E}\left[\left.|\varphi_{ij} - \widehat{\varphi}_{ij}| + |p(y = j|m(x)) - m_j^\theta(x)|\right|y = j, h(x) = i\right]$$
$$= |\varphi_{ij} - \widehat{\varphi}_{ij}| + \mathbb{E}\left[\left.|p(y = j|m(x)) - m_j^\theta(x)|\right|y = j\right]$$

We additionally employ the conditional independence assumption to arrive at the last line.

Plugging this back in to Equation (29), we obtain

$$\mathbb{E}\left[\eta(x, y)\right] \leq \sum_{i=1}^{K}\sum_{j=1}^{K} P(y = j)\varphi_{ij}|\varphi_{ij} - \widehat{\varphi}_{ij}|$$
$$+ \sum_{j=1}^{K} p(y = j)\mathbb{E}\left[\left.|p(y = j|m(x)) - m_j^\theta(x)|\right|y = j\right]$$

Since $\varphi_{ij}, p(y = 1) \leq 1$, the first summand is at most $\sum_{i=1}^{K}\sum_{j=1}^{K}|\varphi_{ij} - \widehat{\varphi}_{ij}| = ||\varphi - \widehat{\varphi}||_1$. In fact, the first summand is typically much smaller than $||\varphi - \widehat{\varphi}||_1$ – for example, if all classes are equally likely, the first summand is at most $\frac{1}{K}||\varphi - \widehat{\varphi}||_1$.

The second summand is readily recognized as the $\ell_1$ marginal calibration error [Kumar et al., 2019]. $\qquad\square$

## Appendix K   Human-Machine Combinations

Although our main focus is on human-model combinations, our method can be applied to the combination of a non-probabilistic model (i.e. that predicts only a label) and a model that outputs a distribution over classes. To demonstrate this, we perform an experiment where we discard the probabilistic information from one of our models (detailed in Section 5 and Appendix F) and treat it as only outputting a label. We then combine the resulting predictions with the other available models for the given dataset. In this section, we call the label-producing model the *hard predictor*, and the distribution-producing model the *soft predictor*.

However, we find that such a combination of two models does not generally improve over the individual components. We hypothesize that this is due to a lack of diversity in the predictions of the models. To investigate this, we compute a simple set of diversity statistics, namely the joint distribution of binary outcomes. That is, we estimate the quantities

$$q_{cc} = \mathbb{P}\left\{y_h = y \wedge y_s = y\right\} \qquad q_{ci} = \mathbb{P}\left\{y_h = y \wedge y_s \neq y\right\}$$
$$q_{ic} = \mathbb{P}\left\{y_h \neq y \wedge y_s = y\right\} \qquad q_{ii} = \mathbb{P}\left\{y_h \neq y \wedge y_s \neq y\right\}$$

where $y_h$ is the label produced by the hard predictor, $y_s$ is the label produced by the soft predictor, and $y$ is the true label. Intuitively, if a combination method always selects either $y_h$ or $y_s$, then the best achievable combination error rate is $q_{ii}$, and so these diversity statistics capture some of the limitations of any such method.

We see in our results below that the diversity of two models is generally less than that of a model and a human, suggesting a partial explanation as to why the combinations of two models tend to under-perform.

| | | Error Rate | | | | | | |
|---|---|---|---|---|---|---|---|---|
| Hard | Soft | Hard | Soft | P+L | $q_{cc}$ | $q_{ii}$ | $q_{ic}$ | $q_{ci}$ |
| ResNet-164 | ResNet-110 | 6.47 | 11.03 | 6.57 | 86.97 | 4.47 | 2.00 | 6.57 |
| Human | — | 4.47 | — | 3.70 | 85.90 | 1.40 | 3.07 | 9.63 |
| PreResnet-164 | — | 5.13 | — | 5.33 | 87.23 | 3.40 | 1.73 | 7.63 |
| Human | — | 4.47 | — | 3.70 | 85.90 | 1.40 | 3.07 | 9.63 |
| DenseNet | — | 3.63 | — | 4.10 | 88.10 | 2.77 | 0.87 | 8.27 |
| Human | — | 4.47 | — | 3.70 | 85.90 | 1.40 | 3.07 | 9.63 |
| ResNet-110 | ResNet-164 | 10.90 | 6.40 | 8.20 | 87.00 | 4.30 | 6.60 | 2.10 |
| Human | — | 4.70 | — | 2.60 | 89.93 | 1.03 | 3.67 | 5.37 |
| PreResnet-164 | — | 4.90 | — | 4.93 | 92.03 | 3.33 | 1.57 | 3.07 |
| Human | — | 4.70 | — | 2.60 | 89.93 | 1.03 | 3.67 | 5.37 |
| DenseNet | — | 3.20 | — | 3.40 | 92.67 | 2.27 | 0.93 | 4.13 |
| Human | — | 4.70 | — | 2.60 | 89.93 | 1.03 | 3.67 | 5.37 |
| ResNet-110 | PreResnet-164 | 11.57 | 5.33 | 7.73 | 86.83 | 3.73 | 7.83 | 1.60 |
| Human | — | 4.77 | — | 2.80 | 91.03 | 1.13 | 3.63 | 4.20 |
| ResNet-164 | — | 6.10 | — | 5.77 | 91.90 | 3.33 | 2.77 | 2.00 |
| Human | — | 4.77 | — | 2.80 | 91.03 | 1.13 | 3.63 | 4.20 |
| DenseNet | — | 3.40 | — | 3.63 | 93.47 | 2.20 | 1.20 | 3.13 |
| Human | — | 4.77 | — | 2.80 | 91.03 | 1.13 | 3.63 | 4.20 |
| ResNet-110 | DenseNet | 11.03 | 3.20 | 4.87 | 87.93 | 2.17 | 8.87 | 1.03 |
| Human | — | 4.57 | — | 2.10 | 92.83 | 0.60 | 3.97 | 2.60 |
| ResNet-164 | — | 5.57 | — | 4.30 | 93.20 | 1.97 | 3.60 | 1.23 |
| Human | — | 4.57 | — | 2.10 | 92.83 | 0.60 | 3.97 | 2.60 |
| PreResnet-164 | — | 5.17 | — | 3.93 | 93.70 | 2.07 | 3.10 | 1.13 |
| Human | — | 4.57 | — | 2.10 | 92.83 | 0.60 | 3.97 | 2.60 |

Table 13: Two model combinations and diversity statistics on CIFAR-10H.

| | | | Error Rate | | | | | | |
|---|---|---|---|---|---|---|---|---|---|
| Noise | Hard | Soft | Hard | Soft | P+L | $q_{cc}$ | $q_{ii}$ | $q_{ic}$ | $q_{ci}$ |
| 80 | Googlenet | VGG19 | 14.67 | 11.41 | 12.06 | 82.36 | 8.44 | 6.23 | 2.97 |
| 80 | Human | — | 10.43 | 11.41 | 6.43 | 81.38 | 3.23 | 7.21 | 8.18 |
| 95 | Googlenet | — | 17.21 | 12.72 | 14.69 | 80.11 | 10.04 | 7.17 | 2.68 |
| 95 | Human | — | 14.52 | 12.72 | 8.67 | 77.26 | 4.50 | 10.02 | 8.22 |
| 110 | Googlenet | — | 19.24 | 15.62 | 16.88 | 77.39 | 12.25 | 6.99 | 3.38 |
| 110 | Human | — | 23.79 | 15.62 | 11.95 | 68.19 | 7.60 | 16.18 | 8.02 |
| 125 | Googlenet | — | 26.75 | 21.73 | 23.63 | 68.63 | 17.11 | 9.64 | 4.62 |
| 125 | Human | — | 38.91 | 21.73 | 19.93 | 52.36 | 13.00 | 25.91 | 8.73 |
| 80 | VGG19 | Googlenet | 11.41 | 14.67 | 11.34 | 82.36 | 8.44 | 2.97 | 6.23 |
| 80 | Human | — | 9.85 | 14.67 | 6.81 | 78.88 | 3.40 | 6.46 | 11.28 |
| 95 | VGG19 | — | 12.72 | 17.21 | 12.69 | 80.11 | 10.04 | 2.68 | 7.17 |
| 95 | Human | — | 12.71 | 17.21 | 9.90 | 75.52 | 5.43 | 7.27 | 11.78 |
| 110 | VGG19 | — | 15.62 | 19.24 | 15.43 | 77.39 | 12.25 | 3.38 | 6.99 |
| 110 | Human | — | 24.93 | 19.24 | 15.65 | 65.36 | 9.52 | 15.41 | 9.72 |
| 125 | VGG19 | — | 21.73 | 26.75 | 21.98 | 68.63 | 17.11 | 4.62 | 9.64 |
| 125 | Human | — | 39.97 | 26.75 | 24.58 | 50.46 | 17.18 | 22.79 | 9.57 |

Table 14: Two model combinations and diversity statistics on ImageNet-16H.

## Appendix References

Thomas B Berrett, Yi Wang, Rina Foygel Barber, and Richard J Samworth. The conditional permutation test for independence while controlling for confounders. *Journal of the Royal Statistical Society: Series B (Statistical Methodology)*, 82(1):175–197, 2020.

Eli Bingham, Jonathan P Chen, Martin Jankowiak, Fritz Obermeyer, Neeraj Pradhan, Theofanis Karaletsos, Rohit Singh, Paul Szerlip, Paul Horsfall, and Noah D Goodman. Pyro: Deep universal probabilistic programming. *The Journal of Machine Learning Research*, 20(1):973–978, 2019.

Charles R. Harris, K. Jarrod Millman, Stéfan J. van der Walt, Ralf Gommers, Pauli Virtanen, David Cournapeau, Eric Wieser, Julian Taylor, Sebastian Berg, Nathaniel J. Smith, Robert Kern, Matti Picus, Stephan Hoyer, Marten H. van Kerkwijk, Matthew Brett, Allan Haldane, Jaime Fernández del Río, Mark Wiebe, Pearu Peterson, Pierre Gérard-Marchant, Kevin Sheppard, Tyler Reddy, Warren Weckesser, Hameer Abbasi, Christoph Gohlke, and Travis E. Oliphant. Array programming with NumPy. *Nature*, 585(7825):357–362, September 2020.

Meelis Kull, Miquel Perello Nieto, Markus Kängsepp, Telmo Silva Filho, Hao Song, and Peter Flach. Beyond temperature scaling: Obtaining well-calibrated multi-class probabilities with dirichlet calibration. In H. Wallach, H. Larochelle, A. Beygelzimer, F. d'Alché-Buc, E. Fox, and R. Garnett, editors, *Advances in Neural Information Processing Systems*, volume 32. Curran Associates, Inc., 2019.

Alexander Marx and Jilles Vreeken. Testing conditional independence on discrete data using stochastic complexity. In *The 22nd International Conference on Artificial Intelligence and Statistics*, pages 496–505. PMLR, 2019.

Sudipto Mukherjee, Himanshu Asnani, and Sreeram Kannan. CCMI: Classifier based conditional mutual information estimation. In *Uncertainty in Artificial Intelligence*, pages 1083–1093. PMLR, 2020.

Radford M Neal et al. MCMC using Hamiltonian dynamics. *Handbook of Markov Chain Monte Carlo*, 2(11):2, 2011.

Adam Paszke, Sam Gross, Francisco Massa, Adam Lerer, James Bradbury, Gregory Chanan, Trevor Killeen, Zeming Lin, Natalia Gimelshein, Luca Antiga, Alban Desmaison, Andreas Kopf, Edward Yang, Zachary DeVito, Martin Raison, Alykhan Tejani, Sasank Chilamkurthy, Benoit Steiner, Lu Fang, Junjie Bai, and Soumith Chintala. Pytorch: An imperative style, high-performance deep learning library. In H. Wallach, H. Larochelle, A. Beygelzimer, F. d'Alché-Buc, E. Fox, and R. Garnett, editors, *Advances in Neural Information Processing Systems 32*, pages 8024–8035. Curran Associates, Inc., 2019.

F. Pedregosa, G. Varoquaux, A. Gramfort, V. Michel, B. Thirion, O. Grisel, M. Blondel, P. Prettenhofer, R. Weiss, V. Dubourg, J. Vanderplas, A. Passos, D. Cournapeau, M. Brucher, M. Perrot, and E. Duchesnay. Scikit-learn: Machine learning in Python. *Journal of Machine Learning Research*, 12:2825–2830, 2011.

Jakob Runge. Conditional independence testing based on a nearest-neighbor estimator of conditional mutual information. In *International Conference on Artificial Intelligence and Statistics*, pages 938–947. PMLR, 2018.