# OpenReview forum: "Combining Human Predictions with Model Probabilities via Confusion Matrices and Calibration"
_NeurIPS.cc/2021/Conference — NeurIPS 2021 Poster_

### Official Review · Reviewer_nC5z · 2021-07-13

**Rating:** 5
**Confidence:** 2

**Summary:**

The authors introduce a simple method for combining predictions from a human and a model (or more generally a non-probabilistic and probabilistic labeler).  The resulting combination model is more accurate than either models or humans alone, and better calibrated than models alone.

**Limitations And Societal Impact:**

seems good

**Main Review:**

Note: I am not familiar with this area of work.

Overall, I think the problem statement is well-motivated and seems to be under-researched.  However, I would've liked to see more thorough experiments on the methods.

I think it would be natural to have baselines where: the model m(x) is finetuned on the labels h(x).  another variant is having m(x) re-trained on ground truth but where it can condition on h(x) for the portion of inputs h(x) is available.  I expect this should be worse in the low data regime but it's less clear in the 5000 datapoint regime.  (It would also be interesting to have humans that get to condition on model predictions, although that is understandably much more difficult to try.)

The combining method at its core is simple, but I would've liked to see ablations on the functional forms for confusion matrix estimation and calibration parameter estimation.  For example, learning K^2 parameters doesn't seem that bad in the 5000 data point regime.  It seems natural to parametrize with a low rank factorization as well.


Other Qs:

- In Figure 2, it's not clear to me why the original CIFAR models aren't already well calibrated on NLL, given that there is no distortion.

- I might have misunderstood, but did Theorem 2 require the conditional independence assumption?  that seems pretty important to state, even if it's kind of obvious after thinking about it."

**Time Spent Reviewing:**

1.5

---

> ### Author Response · Authors · 2021-08-10
> **Author Response to Reviewer nC5z**
>
> Thank you for the review and suggestions on how our work could be improved.
>
> > I think it would be natural to have baselines where: the model m(x) is finetuned on the labels h(x). another variant is having m(x) re-trained on ground truth but where it can condition on h(x) for the portion of inputs h(x) is available.
>
> Approaches where a model is fine-tuned on labels h(x) or conditioned on inputs h(x) is certainly interesting - and we note that there is already existing work in this general direction (that we cite in our paper) on training classification models to adapt to human labels (e.g., [1, 2]). However, we believe that this type of "joint training" is somewhat outside the scope of the problem we are focusing on here in our paper. In particular, we are interested in the problem of combining an existing trained machine classifier with a human labeler. A particular benefit of this approach is that we are agnostic to the structure of the model (other than requiring that it be probabilistic), and hence our method is widely applicable. We also believe that this setup is a very plausible real-world application scenario and worthy of investigation in its own right, for problems where training a model simultaneously with human input is not practical to do, and the only practical option is to combine human and model predictions without retraining the model.
>
> >It would also be interesting to have humans that get to condition on model predictions, although that is understandably much more difficult to try.
>
> We agree that this use case is interesting, namely where the human makes the final decision, but does so with guidance from a machine learning model. However, this setup is considerably different from the one we pursue in our submission. In addition there are no existing datasets (that we are aware of) to use for evaluation of different methods under this scenario. There is  recent work that takes this alternative viewpoint [3] and that we reference in the paper (Line 144).
>
> >The combining method at its core is simple, but I would've liked to see ablations on the functional forms for confusion matrix estimation and calibration parameter estimation. For example, learning K^2 parameters doesn't seem that bad in the 5000 data point regime. It seems natural to parametrize with a low rank factorization as well.
>
> In Appendix C, we provide additional experiments with other calibration methods. In Appendix H, we study additional functional forms for the confusion matrix, namely a single-parameter version ("SP") and a logistic regression based model ("LR"). Given that our method performs well in the limited data regime even with no reductions to the admissible functional form of the confusion matrix, we did not feel that it was necessary to make approximations here. However, we do note that such approximations may be useful in applications with a much larger number of classes K than what we study here.
>
> >In Figure 2, it's not clear to me why the original CIFAR models aren't already well calibrated on NLL, given that there is no distortion.
>
> We assume that you are referring to Table 2 here. In Table 2, we evaluate various calibration metrics (ECE, cwECE, NLL) on a held-out validation set (see line 265-268). The fact that the models have a large validation NLL is simply a reflection of the well-known phenomenon that these models are not well-calibrated after training [4].
>
> > I might have misunderstood, but did Theorem 2 require the conditional independence assumption? that seems pretty important to state, even if it's kind of obvious after thinking about it.
>
> You are correct, Theorem 2 requires the conditional independence assumption. Since this is a statement about our P+L method, the CI assumption is implicit (as CI is used to derive the P+L method). However, we agree that the statement of Theorem 2 could be more explicit, and hence we have updated the phrasing to include "Under the CI Assumption, in expectation. . ." on line 320.
>
> [1] Hussein Mozannar and David Sontag. Consistent Estimators for Learning to Defer to an Expert, ICML 2020.
>
> [2] Bryan Wilder, Eric Horvitz and Ece Kamar. Learning to Complement Humans, IJCAI 2020.
>
> [3] Gagan Bansal, Besmira Nushi, Ece Kamar, Eric Horvitz and Daniel Weld. Optimizing AI for Teamwork, arXiv preprint arXiv:2004.13102
>
> [4] Chuan Guo, Geoff Pleiss, Yu Sun, and Kilian Q. Weinberger. On Calibration of Modern Neural Networks, ICML 2017.

---

### Official Review · Reviewer_X6gK · 2021-07-14

**Rating:** 7
**Confidence:** 3

**Summary:**

The paper "Combining Human Predictions with Model Probabilities via Confusion Matrices and Calibration" proposes a method to obtain a well-performing ensemble model of both an ML classifier (which outputs a probability for every class) and a human observer (who outputs a categorical decision). This is a setting that traditional ensemble methods do not account for; e.g. majority vote would not be possible here since there are only two decision makers in total (1 model, 1 human). The problem may not be common in standard ML but since ML applications are increasingly being used in real-world settings, there are areas where this approach can be applied (such as high-stakes medical decisions). Combining model and human predictions is shown to lead to lower error rates than either of the two alone.

**Limitations And Societal Impact:**

In Section 7 of their paper, the authors discuss limitations and potential societal impact. In particular, I appreciated that the authors took the time to think about potential _negative_ societal consequences as well, which can often be difficult at first (especially for standard "pure research" papers). Well done!


**Main Review:**

## Originality

According to the authors (lines 177ff) , this is the first method that combines a hard label prediction with a probabilistic label prediction. The related work section (Section 3) references relevant prior work, and clearly delineates how the paper differs from existing publications.

Minor comment: The authors write that "This work emphasizes the point that combinations of models that have diversity in how they make predictions can systematically outperform a single model." -> I'm not convinced that this point really needs emphasis, it's rather well-known that correlated predictors are bad, right? (e.g. multicollinearity in regression).


## Quality

The paper uses a nice combination of experimental and theoretical methods, and the results are presented clearly in text, figures and tables. Disclaimer: I am not an expert in this area. That being said, the results (e.g. lines 55f) are convincing despite the simplicity of the approach - for instance, if the achieved accuracy gain were the result of an architectural modification, it would be considered a significant improvement. Of course, this comes at the cost of obtaining human decisions, which is not feasible or practical in many cases, but for specific use cases described by the authors (e.g. medical diagnostics) it may well be useful.

Equation 2 assumes conditional independence, which is an assumption that typically does not hold in practice, at least not completely (see Table 3 in the appendix) but the numerical results are good in spite of a potential violation of this assumption, so I don't consider this to be a big problem.

While the authors focus on the setting where human and machine decisions are combined, their approach is generally applicable to any two decision makers whenever there is a combination of two models, one with confidence scores for all classes and one without (i.e. only outputting a single class). In order to increase the potential appeal and applicability of the paper, it might be helpful to additionally showcase one example where instead of combining human and machine predictions, two machine predictions are combined in this way. This might be an interesting setting e.g. for cases where there is a proprietary model for which only the output decision (but not the confidence) is known, and it would additionally be a nice example for settings where there might be more than a handful (16) of classes.


## Clarity

The introduction is well written and clearly motivates the problem. The paper is well organized and structured.

While the authors perform multiple experiments testing calibration properties, it would be helpful to explicitly state the limitations of the approach: e.g. are there any limitations on the accuracy difference between the two decision makers that need to be met? At which point of poor calibration does the combined model perform worse than e.g. a human observer alone? How "different" / uncorrelated / ... do model and human predictions need to be?

Two minor comments regarding clarity:

- Figure 1 right: it would be helpful to mention the datasets below the two bar plot groups (I'm assuming the left one refers to CIFAR-H and the right one to ImageNet-16H?).

- I was confused about the sentence in line 98: why does a single observer have a confusion matrix? Is the observer shown the image multiple times and then decides differently? Or is this an aggregated confusion matrix over multiple images? Generally, how can you parameterise a probability with an entire confusion matrix? This became a bit more clear when looking at equation (3) but it would be helpful to make the entire paragraph starting with line 98 more precise / self-contained. I'm still not entirely sure about how the values for the confusion matrix are obtained.

- In equation (3), "i" is defined twice: once on the left-hand side of the equation and also again on the right-hand side as the sum counter; this is confusing

#### Code clarity

I appreciate that code was provided. A README would go a long way in making the code more accessible; currently it's mostly a list of files and scripts. The 20+ TODOs don't really help in providing confidence in the code quality: "Do these edge cases really need to be separated out??", "This isn't the right file for this", ... plus, as a suggestion for future code submissions, I would make sure that hard-coded paths are properly anonymized.

## Significance

As explained above, the paper addresses a relevant topic, more precisely, how human and model decisions may be combined. This is probably less relevant for tasks like the investigated one (such as standard ImageNet object recognition), but might become very relevant in the context of medical imaging (where peak performance truly matters, and spending the time of obtaining a human classification decision is a fair price to pay for improved performance).

## MISC

- it would be helpful to use a citation style etc. where one can click on a number in the paper and directly end up at the reference (currently this is not possible, so looking up a reference involves a lot of scrolling down)
- typo line 39: remove "being"
- humans are refered to as "non-probabilistic labelers"; perhaps a footnote could be added to explain that this refers to the fact that humans output a category instead of a probability vector. The term "non-probabilistic labeler" could lead to confusion if understood in the sense that humans would be deterministic and always choose the same label, i.e. that humans wouldn't have internal noise, which is not the case.
- typo line 146: remove "that"
- Table 2: for readers not familiar with the metrics, indicate whether lower=better or higher=better (and perhaps make the leading entries bold?)
- the paper sometimes refers to "human labelers", which may be confusing since usually "labelers" are those who generate labels for the dataset (rather than classify images). Perhaps rephrase accordingly?

**Time Spent Reviewing:**

2h 40min

---

> ### Author Response · Authors · 2021-08-10
> **Author Response to Reviewer X6gK**
>
> We thank the reviewer for the thorough review and thoughtful questions. The various typos pointed out have been fixed (Equation (3), line 39, line 146), and we additionally will be sure to make our citations work as links to the references (they work in our local copy of the paper but not on OpenReview for some reason).
>
> > The authors write that "This work emphasizes the point that combinations of models that have diversity in how they make predictions can systematically outperform a single model." -> I'm not convinced that this point really needs emphasis, it's rather well-known that correlated predictors are bad, right? (e.g. multicollinearity in regression).
>
> Yes, we agree that this is well-known. We note (in case it wasn’t clear) that “this work” is in reference to previous work in this area, and referring to our submission. We will update the camera-ready version (if accepted) to read "This existing line of work emphasizes..." in order to make this more clear.  We believe such a statement is useful, particularly in the introductory paragraphs, for readers who may be new to the field (e.g., new graduate students who might not be fully aware of previous work on this topic) .
>
> > In order to increase the potential appeal and applicability of the paper, it might be helpful to additionally showcase one example where instead of combining human and machine predictions, two machine predictions are combined in this way.
>
> This is a good suggestion and an interesting use-case. Although we do mention this possibility in lines 87-90, we agree that an additional experiment showcasing this scenario would be beneficial. If accepted, we will include in our camera-ready submission at least one additional experiment combining a probabilistic model (i.e. that outputs a probability distribution over classes) and a non-probabilistic model (i.e. that outputs a categorical prediction).
>
> > While the authors perform multiple experiments testing calibration properties, it would be helpful to explicitly state the limitations of the approach: e.g. . .
>
> This is an interesting point and we agree that these are important questions to explore - however we believe they are beyond the scope (and space limitations) of this present paper.  In fact, in ongoing (but currently unpublished) parallel work we are investigating these types of questions. Our preliminary results suggest that it is indeed possible to analyze the limits of combining humans and models, where the accuracy of, and correlation between, the predictors plays a key role.
>
> > Minor comments regarding clarity...
> >> Figure 1 right
>
> We will update Figure 1 to include the dataset names (CIFAR-10H on the left, ImageNet-16H on the right).
>
> >> Line 98
>
> The confusion matrix for a human is indeed obtained by aggregating over multiple images. For example, the maximum likelihood estimate for $\varphi_{ij}$ is (up to normalization) the number of times a human predicted label $i$ when the ground-truth label was $j$, counted across all such labeled images. In Section 4, we additionally detail a Bayesian procedure for estimating the entries of $\varphi$, allowing us to incorporate informative prior information.
>
> Our camera-ready version (if accepted) will rephrase the paragraph starting at line 98 to emphasize the point that the term $p(h(x) | y)$ is modeled by the entries $\varphi_{ij}$ for particular values of $h(x) = i$ and $y = j$, in order to make the setup more clear.
>
> >> Code clarity
>
> Thank you for taking the time to evaluate our code, we appreciate the effort. We have updated our code submission to be fully anonymized and more accessible.
>
> > Misc.
> >> Non-Probabilistic Labelers
>
> We agree that this term could be defined more clearly for the reader. Our draft has been updated to emphasize this distinction. In particular, line 88 now reads "... combinations of a non-probabilistic labeler (whose output is categorical) and a probabilistic labeler (whose output is a distribution over classes)."
>
> >>Table 2
>
> In all cases, lower is better. We have updated the caption of Table 2 to reflect this ("For all metrics, lower is better.") However, we would prefer not to bold entries in this table, as we are not suggesting that using 5000 datapoints for calibration is desirable. Instead, the main points we are trying to make here are that (i) combinations tend to be better calibrated than the model alone (comparing the paired columns), and (ii) most of the recalibration benefits can be had with very small amounts of labeled data (comparing the "10 Datapoints" and "5000 Datapoints" columns).
>
> >> "Human Labelers"
>
> Throughout, we refer to the "true" labels for an input as ground truth, and we explicitly note that the "human labeler" may be noisy relative to ground truth (e.g. lines 81-83).

---

> > ### Comment · Reviewer_X6gK · 2021-08-11
> > **Thanks & justification of final score**
> >
> > I would like to thank the authors for their detailed response.
> >
> > The authors have fully addressed my suggestions regarding clarity, which I appreciate.
> >
> > The authors also replied to my two more substantive suggestions, but I wasn't 100% convinced by their response:
> > 1. "showcase one example where instead of combining human and machine predictions, two machine predictions are combined in this way."
> > -> The authors pledge to include such an example in the final version; from a reviewer's perspective it would have been even better to provide the results already instead of referring to a future version of this paper.
> >
> > 2. "state the limitations of the approach: e.g. are there any limitations on the accuracy difference between the two decision makers that need to be met? At which point of poor calibration does the combined model perform worse than e.g. a human observer alone? How "different" / uncorrelated / ... do model and human predictions need to be?"
> > -> The authors write that this is out of the scope of their paper. I don't think I agree that it is helpful to consider analyzing the limitations of one's approach out of scope; e.g. investigating at which point of poor calibration a combined model performs worse than e.g. a human observer alone could have been done with a brief experiment described in the appendix. Nonetheless, the question of scope is in the end a question that the authors have to decide, not a reviewer.
> >
> > That being said, I am still convinced this is a solid submission, and I highly recommend acceptance. Having read the other reviewer's comments, and the respective author responses, as well as the author's response to my suggestions, I have decided to keep my initial "7: Good paper, accept" rating.

---

### Official Review · Reviewer_NHxx · 2021-07-16

**Rating:** 7
**Confidence:** 4

**Summary:**

The current work focuses on the interesting prospect and problem of combining human and machine classification information per input, especially in the small-data regime, for the sake of ultimately improving accuracy. The authors take a probabilistic approach to motivating their strategy for combining human-machine information, leveraging a carefully chosen prior to enable inference of a reliable human confusion matrix when the number of annotators is low. They ultimately find that the resulting combination (evaluated in the context of popular image classification tasks) improves accuracy over both humans and models alone, and with drastically less data than previous work. They show more formally that this success is in part driven by both the accuracy of humans/models as well as the confidence values of each.

**Limitations And Societal Impact:**

The authors discuss plausible possible negative impacts that I did not foresee myself.

**Main Review:**

To my knowledge, the methods employed, formal arguments, and conclusions are generally valid. The paper is atypically clear in explaining its aims, proposal, and evaluation. While I expect that these results will be robust, I would have preferred that the authors discuss them in the context of past work as well as different interpretation of the human behavioral datasets.

For example, the CIFAR-10H dataset contains over 500,000 judgements spanning thousands of annotators having viewed different subsets of images. Under the assumption that human annotators are noisy labelers with similar classification behavior, each labeler provides a sample from a "mental class-conditional distribution". However, if classification behavior is diverse, the resulting labels are more like argmax predictions from an ensemble of humans. In the former case, we can think of this sort of data as sort of representing a model, whereas in the latter, we might instead think of it as representing an ensemble of models in itself. The implications for the proposed method and its applicability given these two interpretations is not clear. Since the authors sample from single annotators, I think this is worth discussing.

Moreover, the original CIFAR-10H work performed a sort of validation of the soft labels by evaluating models trained on them on out-of-distribution datasets and against adversarial attacks. This was effectively a kind of argument that the confidence information conveyed in this sort of data is useful for classification. Given these results, why shouldn't we expect this confidence information to be useful in the current work? Further, why wouldn't the authors evaluate their hybrid method using OOD datasets, where accuracy on validation sets can be misleading as a target of combination?

Nonetheless, I think the work provides a unique contribution, and provides a good reason to be optimistic about leveraging these sorts of human datasets without having to scale dataset collection by a factor of n_annotators_per_image in a way that is prohibitively expensive.

**Time Spent Reviewing:**

4

---

> ### Author Response · Authors · 2021-08-10
> **Author Response to Reviewer NHxx**
>
> We thank the reviewer for detailed comments and thoughtful questions.
>
> > While I expect that these results will be robust, I would have preferred that the authors discuss them in the context of past work as well as different interpretation of the human behavioral datasets. . . The implications for the proposed method and its applicability given these two interpretations is not clear. Since the authors sample from single annotators, I think this is worth discussing.
>
> Thanks for these comments. We agree that our experiments, where we randomly sample labelers, are in effect equivalent to using an ensemble (or mixture) of individual human “models.” A similar point was also raised by Reviewer YY6P. Please see our response to that reviewer, where we detail an additional experiment where we fit a combination model to each individual labeler in our datasets. In summary, we show that the performance of our method is robust to the choice of selection method, obtaining comparable performance when we fit a contusion matrix to each individual (as in our response to Reviewer YY6P) or when we sample from individuals (as we do in our submission).
>
>
> > Moreover, the original CIFAR-10H work performed a sort of validation of the soft labels by evaluating models trained on them on out-of-distribution datasets and against adversarial attacks. This was effectively a kind of argument that the confidence information conveyed in this sort of data is useful for classification.
> >> Why shouldn't we expect this confidence information to be useful in the current work?
>
> We agree that the distribution of human labels for a single input could potentially be useful for increasing performance. However, such confidence information requires multiple individual humans to label a single input, which will typically be infeasible in many practical situations, e.g. in domains such as medical imaging where human-generated labels are expensive to obtain. For this reason, we focus on the scenario where only a single human-generated label is available per input.
>
> >> Further, why wouldn't the authors evaluate their hybrid method using OOD datasets, where accuracy on validation sets can be misleading as a target of combination?
>
> This is potentially interesting to consider, but in this work we focus on evaluating our methodology on the standard train-test setup for classification datasets. On an OOD dataset, we would in general expect to see a drop in performance, as the machine classifier would likely not only be less accurate but also worse calibrated [1]. While human-machine combinations that are robust to distribution shifts are of clear practical importance, we restrict our attention in this work to the i.i.d. setting, where we are able to establish our method as a strong baseline which future work (possibly investigating the OOD case) can build on.
>
>
> [1] Yaniv Ovadia et al. Can You Trust Your Model’s Uncertainty? Evaluating
> Predictive Uncertainty Under Dataset Shift, NeurIPS 2019.

---

> > ### Comment · Reviewer_NHxx · 2021-08-31
> > **no change to score**
> >
> > After reading the other reviews and responses, I am still in favor of acceptance.

---

### Official Review · Reviewer_YY6P · 2021-08-01

**Rating:** 8
**Confidence:** 4

**Summary:**

This paper presents and investigates methods for combining human and machine predictions. In the proposed setup, such a combination relies on instance-level probability outputs from a machine learning model and class-level information from human annotators. In a series of extensive experiments, the authors show that their method, which relies on model recalibration and confusion matrices, outperforms both humans and ML models. To support the strong empirical results, the authors demonstrate theoretically how this combination gains from the recalibration process.

The main contributions of this paper are its simple and sound methodology, its strong experimental analysis and its useful theoretical grounding. It is also clearly written and insightful.


**Limitations And Societal Impact:**

This paper can potentially have a substantial positive societal impact, as successfully combining human and machine predictions can improve decisions made in high-stakes settings. Moreover, better-understanding the effect of model calibration on downstream decisions supported by machine learning models can surely improve their safe dissemination in the wild. I believe that the authors have properly explained the benefits in the context of high-stakes applications, though more emphasis can be made on the importance of understanding model calibration.

I also think that the authors have adequately addressed the limitations of their work, which are the focus on computer vision applications and their reliance on the conditional independence assumption.


**Main Review:**

This is a strong, clearly written paper that improves our understanding of an important subject. While it is not totally original in its approach (there is a plethora of existing work on combining humans and machines for better decision making), it is extremely clear in its approach and it makes several important contributions to this literature.

Mainly, the proposed framework generalizes existing approaches to human-machine combinations and can be used in the future for other applications and with other algorithmic solutions. Second, the empirical analysis sheds light on the sensitivity of such approaches to model calibration, and specifically to overconfidence. Deep neural networks are notoriously over confident, and understanding how this affects their usefulness as decision support systems is crucial. Finally, the theoretical analysis is insightful and relevant, shedding light on the sensitivity to model overconfidence.

However, while I appreciate the authors’ acknowledgement of the paper’s limitations, I do think that more can be made to address them. First, the reliance on conditional independence is indeed problematic and might result in the approach being less useful when estimating conditional independence is infeasible. Second, it would be interesting to see the performance of the approach on datasets which do not involve image classification. Specifically, extending to NLP, where overconfidence is even more of a problem, would also be interesting. Third, it might be interesting to explore more advanced recalibration methods, or optimize for calibration in the first place (for example see Kumar et al 2018).

Finally, it is unclear whether the results are robust to other human selection methods. The authors choose humans randomly, but it is unclear if their method is sensitive to cases where each human annotates many examples and is biased towards specific types of decisions. I might have missed that in the paper, so perhaps better explaining the sensitivity to the human selection process in the CIFAR and ImageNet datasets might be helpful.


**Time Spent Reviewing:**

3

---

> ### Author Response · Authors · 2021-08-10
> **Author Response to Reviewer YY6P**
>
> We thank the reviewer for their detailed comments and for the encouraging review.
>
> > The reliance on conditional independence is indeed problematic and might result in the approach being less useful when estimating conditional independence is infeasible.
> > Extending to NLP, where overconfidence is even more of a problem, would also be interesting.
>
> We agree with these sentiments, and point to these in our Limitations section. Developing robust methodologies that relax the conditional independence assumption is certainly a useful direction for future work on this topic. In practice, one might expect diminishing returns, where potential reductions in bias (with more flexible models) could be offset by increases in variance, particularly when human labels are scarce and the number of classes is large.
>
> In terms of empirically evaluating our approach on non-image domains, we were limited in our work by the availability of datasets with large high-quality sets of human annotations relative to some established ground truth. Such data is largely limited to image datasets like CIFAR-10H and Imagenet at the moment - evaluating our approach on non-image datasets, such as NLP, is something that we (or others) will hopefully be able to do in the future once suitable annotated datasets become available.
>
> > It might be interesting to explore more advanced recalibration methods, or optimize for calibration in the first place.
>
> In Appendix C, we provide additional experimental results with calibration methods beyond temperature scaling. In particular, we experiment with ensemble temperature scaling [1] and I-max binning [2]. Our results suggest that in some cases, using a more powerful calibration method can result in performance gains, but we focus in the main part of the paper on temperature scaling given its simplicity and competitive performance [2, 3]. Moreover, the simplicity of temperature scaling makes specifying informative priors viable, which is a key ingredient in the label efficiency of our approach. With regards to optimizing for calibration, we note that such approaches can possibly be used in conjunction with our method -- however, the benefit is not clear, as post-hoc calibration is already an integral part of our approach.
>
> > Finally, it is unclear whether the results are robust to other human selection methods. The authors choose humans randomly, but it is unclear if their method is sensitive to cases where each human annotates many examples and is biased towards specific types of decisions.
>
> This is an insightful observation that Reviewer NHxx also points out. After discussing this idea, we conjectured that the P+L method using data from individual labelers should perform at least as well as random sampling of labelers since the method would be able to leverage individual-specific labeling information. In addition, given the results in Figure 2, showing significant improvements with as few as 10 human labels, the method should be robust to having relatively few labels per individual.
>
> To investigate this conjecture we conducted the following experiments. We fit a combination model (P+L, using MAP inference and temperature scaling as detailed in Section 4) separately to each individual human labeler in our datasets (CIFAR-10H, ImageNet-16H) using 25 labeled datapoints per individual. We then evaluated the resulting accuracy of the P+L combination on the remaining held-out datapoints for each individual (per individual, ~175 images on CIFAR-10H, ~25 images on ImageNet-16H). We repeated this experiment for ten random train/test splits.
>
> Below, we report the mean error rates ($\pm$ one standard deviation), averaged over all individuals and train/test splits ("P+L (Individual)"). We note that the human/model error rates vary slightly from the reported numbers in our paper due to different random train/test splits. In general, the results support our conjecture that fitting a confusion matrix to each individual results in performance comparable to the approach that samples random individuals (as in our submission -- "P+L (Sampled)" from tables 4/5/6 in Appendix C).
>
> ### CIFAR-10H:
>
> | Model Name    | Human          | Model          | P+L (Individual) | P+L (Sampled) |
> |---------------|----------------|----------------|------------------|------------------|
> | ResNet-110    | $5.11 \pm 5.7$ | $11.1 \pm 2.5$ | $4.06 \pm 1.76$  | $3.83 \pm 0.15$  |
> | ResNet-164    | &mdash;        | $6.09 \pm 1.8$ | $2.87 \pm 1.4$   | $2.78 \pm 0.15$  |
> | PreResNet-164 | &mdash;        | $4.92 \pm 1.6$ | $2.62 \pm 1.3$   | $2.43 \pm 0.22$  |
> | DenseNet-BC   | &mdash;        | $3.31 \pm 1.4$ | $2.04 \pm 1.1$   | $2.01 \pm 0.15$  |
>
> ### VGG-19 on ImageNet-16H:
>
> | Noise Level | Human            | Model           | P+L (Individual) | P+L (Sampled) |
> |-------------|------------------|-----------------|------------------|------------------|
> | 80          | $9.81 \pm 8.5$   | $11.29 \pm 6.3$ | $6.15 \pm 4.9$   | $6.03 \pm 0.5$   |
> | 95          | $13.87 \pm 9.7$  | $12.56 \pm 6.3$ | $8.11 \pm 5.5$   | $7.89 \pm 0.4$   |
> | 110         | $23.13 \pm 12.3$ | $15.49 \pm 7.6$ | $11.59 \pm 6.7$  | $11.62 \pm 0.5$  |
> | 125         | $39.84 \pm 13.4$ | $21.92 \pm 8.2$ | $20.87 \pm 8.3$  | $19.63 \pm 0.7$  |
>
> ### DenseNet-BC on ImageNet-16H:
>
>  | Noise Level | Human            | Model           | P+L (Individual) | P+L (Sampled) |
> |-------------|------------------|-----------------|------------------|------------------|
> | 80          | $9.81 \pm 8.5$   | $14.49 \pm 7.0$ | $7.11 \pm 5.6$   | $6.80 \pm 0.47$  |
> | 95          | $13.87 \pm 9.7$  | $17.16 \pm 7.0$ | $9.72 \pm 6.4$   | $9.67 \pm 0.40$  |
> | 110         | $23.13 \pm 12.3$ | $18.88 \pm 8.2$ | $14.25 \pm 7.7$  | $14.09 \pm 0.55$ |
> | 125         | $39.84 \pm 13.4$ | $26.63 \pm 9.2$ | $24.30 \pm 9.3$  | $22.60 \pm 0.63$ |
>
> Note that for our larger-scale experiments (i.e., the main experiments we report on in Section 5) we need large numbers of labels (e.g., for the learning curves in Figure 2) so we can't replace our experiments with the random-sampling approach with experiments using individual labelers throughout the paper.
>
> If accepted, we will briefly discuss these individual-level findings, and how they relate to our methodology, in the main text of our camera-ready paper and will include the details of the experimental setup and results in an additional Appendix section.
>
>
> [1] Jize Zhang, Bhavya Kailkhura, and T. Yong-Jin Han. Mix-n-Match: Ensemble and Compositional Methods for Uncertainty Calibration in Deep Learning, NeurIPS 2020.
>
> [2] Kanil Patel, William Beluch, Bin Yang, Michael Pfeiffer, and Dan Zhang. Multi-Class Uncertainty Calibration via Mutual Information Maximization-based Binning, arXiv preprint arXiv:2006.13092.
>
> [3] Jonathan Wenger, Hedvig Kjellström, and Rudolph Triebel. Non-Parametric Calibration for Classification, AISTATS 2020.

---

> > ### Comment · Reviewer_YY6P · 2021-08-26
> > **Comment**
> >
> > I wish to thank the authors for the extremely insightful response and for the additional experiments conducted.
> > Most of my concerns were addressed and I would be happy to see the paper in the main conference.

---

### Decision · Program_Chairs · 2021-09-27

**Decision:**

Accept (Poster)

**Comment:**

Strengths:
- Simple, well-executed approach
- Clever combination of machine and human strengths (probabilistic vs categorical predictions)
- Clear idea, clear writing

Weaknesses:
- Some unclarity regarding robustness of empirical results
- Needs discussion regarding individual vs. collective human-labeling behavior
- Authors should be more precise about limitations, and address them

Summary:

After some helpful discussions between the reviewers and the authors (including detailed responses by the authors, which the reviewers appreciated), reviewers are mostly in agreement that this is a strong submission which should be of interest to the community. Concerns mostly regarded the experimental setup. Here, reviewers were concerned about whether the empirical results would prove to be results under reasonable variations of the setup. Relatedly, reviewers also pointed out that results could be specific to how the authors chose to select labelers (at random), and asked the authors to clarify their reasoning behind this and they way in which they present their results in light of this experimental design decision. Authors are encouraged to clarify the issues regarding diversity in individual user behavior vs. collective user behavior, and how it relates to their experiments.
Finally, two reviewers were glad to see that the authors acknowledged some of the limitations of their approach, but at the same time, urged the authors to discuss them with more precision, and – more importantly – take steps to address them in the final version of the paper.